# Lipid-mediated PX-BAR domain recruitment couples local membrane constriction to endocytic vesicle fission

Johannes Schöneberg[1,*,†], Martin Lehmann[2,3,*], Alexander Ullrich[1], York Posor[2,3], Wen-Ting Lo[2], Gregor Lichtner[1,2], Jan Schmoranzer[2,3,†], Volker Haucke[2,3,4] & Frank Noé[1]

Clathrin-mediated endocytosis (CME) involves membrane-associated scaffolds of the bin-amphiphysin-rvs (BAR) domain protein family as well as the GTPase dynamin, and is accompanied and perhaps triggered by changes in local lipid composition. How protein recruitment, scaffold assembly and membrane deformation is spatiotemporally controlled and coupled to fission is poorly understood. We show by computational modelling and super-resolution imaging that phosphatidylinositol 3,4-bisphosphate [PI(3,4)P$_2$] synthesis within the clathrin-coated area of endocytic intermediates triggers selective recruitment of the PX-BAR domain protein SNX9, as a result of complex interactions of endocytic proteins competing for phospholipids. The specific architecture induces positioning of SNX9 at the invagination neck where its self-assembly regulates membrane constriction, thereby providing a template for dynamin fission. These data explain how lipid conversion at endocytic pits couples local membrane constriction to fission. Our work demonstrates how computational modelling and super-resolution imaging can be combined to unravel function and mechanisms of complex cellular processes.

[1] Department of Mathematics and Computer Science, Freie Universität Berlin, Berlin 14195, Germany. [2] Leibniz-Institut für Molekulare Pharmakologie, Robert-Roessle-Straße 10, Berlin 13125, Germany. [3] Faculty of Biology, Chemistry, Pharmacy, Freie Universität Berlin, Berlin 14195, Germany. [4] NeuroCure Cluster of Excellence, Charité Universitätsmedizin Berlin, Virchowweg 6, Berlin 10117, Germany. * These authors contributed equally to this work. † Present addresses: Department of Molecular and Cell Biology, University of California, Berkeley, Berkeley, California 94720, USA (J.Schö.); Charité Universitätsmedizin Berlin, Virchowweg 6, 10117 Berlin, Germany (J.Schm.). Correspondence and requests for materials should be addressed to J.S. (email: schmoranzer@charite.de) or to V.H. (email: haucke@fmp-berlin.de) or to F.N. (email: frank.noe@fu-berlin.de).

Dynamic membrane remodelling by membrane fission and fusion[1] is essential for nearly all cell physiological functions including secretion, cell signalling, migration and development, among many other processes[2,3]. A prominent example of membrane fission is the internalization of parts of the plasma membrane by clathrin-mediated endocytosis (CME)[4]. CME is initiated at phosphatidylinositol-4,5-bisphosphate [PI(4,5)$P_2$]-enriched plasma membrane sites[5] by the recruitment of early-acting endocytic factors such as epidermal growth factor receptor substrate 15 (Eps15)[6], intersectin[7,8], AP-2 (refs 9,10), or clathrin assembly lymphoid myeloid leukaemia (CALM)[11,12], as well as the bin-amphiphysin-rvs (BAR)[13–15] domain-containing protein scaffold FCH domain only 1 and 2 (FCHo1/2)[6,8,16]. These factors sort endocytic cargo (for example, receptors, channels and so on destined for internalization) and generate a membrane template for the assembly of the clathrin coat[17–20]. During the following maturation process, endocytic CCPs undergo massive remodelling involving local membrane deformation and constriction of the CCP neck[17] to provide a template for fission mediated by dynamin[3,21] and aided by coordinated actin polymerization[22–24]. Studies in yeast[25] and mammalian cells suggest that endocytic BAR domain proteins with different curvatures contribute to endocytic membrane remodelling by sensing and/or inducing membrane curvature and facilitate membrane invagination and scission[13–15,26,27]. In many cases, however, their precise function in the endocytic process remains poorly understood.

Endocytic membrane remodelling is accompanied by and possibly driven by changes in the local lipid composition. PI(4,5)$P_2$ is hydrolysed as CME progresses[28], while CCP maturation is accompanied by the clathrin-mediated recruitment of the phosphatidylinositol 3,4-bisphosphate [PI(3,4)$P_2$]-synthesizing enzyme class II PI 3-kinase C2α (PI3KC2α)[29], prior to dynamin-mediated fission. This suggests that endocytic membrane remodelling during CCP maturation requires phosphoinositide (PI) lipid conversion from PI(4,5)$P_2$ to PI(3,4)$P_2$ (ref. 29). How this lipid-based mechanism couples membrane constriction to membrane fission is unknown. In vitro studies have found that helical assemblies of multiple dynamin rings can mediate constriction and fission of large liposomal membrane templates[13,21,30,31], while recent work using physiological PI levels shows that dynamic scaffold assembly depends on the presence of a constricted membrane template[32], suggesting that additional factors may be required to provide such a template for dynamin-mediated constriction and fission in vivo. Moreover, model membranes lack stabilizing proteins and are, thus, more flexible than the plasma membrane in living cells, where dynamin associates with highly curved membranes of about 20 nm diameter corresponding to its preferred membrane template in vitro[33,34]. When bound to such a short highly constricted membrane neck, a single dynamin ring is sufficient to catalyse fission of invaginated CCPs[18,35].

We therefore hypothesized that other factors, such as dynamin-binding BAR domain proteins, are recruited by PI(3,4)$P_2$ and mediate or facilitate the initial constriction of the membrane underlying the transition from U-shaped to Ω-shaped CCPs that are the substrate for membrane fission by dynamin in vivo. Here we show by computational modelling and super-resolution imaging that PI(3,4)$P_2$ synthesis within the clathrin-coated area of endocytic intermediates triggers selective recruitment of SNX9, a PX-BAR domain protein that functions in the coordination of membrane remodelling and fission via interactions with actin-regulating and endocytic proteins[36,37], to the invagination neck, where it couples local membrane constriction to endocytic vesicle fission by dynamin.

## Results

**A model for lipid-driven endocytic protein dynamics.** Cell biological and genetic data suggest that endocytic membrane remodelling during CCP maturation may require partial PI conversion from PI(4,5)$P_2$ to PI(3,4)$P_2$ (refs 13,38), a process that depends on the local synthesis of PI(3,4)$P_2$ by PI3KC2α (ref. 29). PI(3,4)$P_2$ is thought to recruit dynamin-binding BAR domain proteins such as SNX9 (and its close paralogue SNX18)[29] to mediate the initial constriction of the membrane underlying the transition from U-shaped to Ω-shaped CCPs that are the substrate for membrane fission by dynamin in vivo.

The local PI(3,4)$P_2$ concentration at CCPs that may serve to recruit endocytic proteins including BAR domain proteins such as SNX9 and SNX18 in CME depends on the activity of PI3KC2α, which produces this lipid. Local production of PI(3,4)$P_2$ by PI3KC2α within the protein-dense environment of a CCP raises the question if, given PI(3,4)$P_2$ production and diffusion rates, a local PI(3,4)$P_2$ gradient can be generated and maintained. Moreover, if such a lipid gradient were to exist, how would it influence the recruitment of PI-binding endocytic proteins such as SNX9 to CCPs?

To address these basic questions, we designed a computational reaction-diffusion master-equation model[39,40] that captures the complex interplay between PI lipids and PI-binding endocytic proteins in an assembled 2-dimensional clathrin lattice (Fig. 1a; Supplementary Table 1). The model describes the locations and diffusional motion of all lipid and protein copies involved, as well as lipid production and protein-ligand binding/dissociation events. The size of the model clathrin lattice corresponds to a coated vesicle of 150 nm in diameter (Fig. 1a; Supplementary Table 1), consistent with the outer diameter of invaginated Ω-shaped CCPs and coated vesicles in non-neuronal cells that we determined by electron microscopy (161 ± 20 nm, $n = 30$). Membrane binding of proteins is treated by a model, which describes both, the non-specific binding affinity of proteins to the membrane (for example, due to charge complementarity) and specific binding of PI lipids to each of the protein's PI-binding domains (Fig. 1b–d). The binding model parameters were determined for all endocytic proteins used here by fitting to experimentally known binding constants (Supplementary Table 1). Since the PI-binding affinity has not been available for the dynamin-associated PI-binding PX-BAR and SH3 domain protein SNX9, a candidate membrane remodelling PI(3,4)$P_2$ effector[29], we determined it here experimentally. Surface plasmon resonance (SPR) measurements revealed that purified recombinant SNX9 (Fig. 1e) binds to PI(3,4)$P_2$ and PI(4,5)$P_2$ with nearly equal affinity of about 1–1.5 μM (Fig. 1f; Supplementary Fig. 1), consistent with prior qualitative data[29,41].

Protein copy numbers determined by quantitative proteomics[42] were used as a reference for most endocytic proteins (Supplementary Table 1). The average copy number of PI3KC2α and SNX9 at CCPs were determined using genome-engineered SK-MEL-2 cells expressing mRFP-clathrin light chain (mRFP-CLC) with its endogenous locus[35]. Previous data using genome-engineered SK-MEL-2 cells co-expressing dynamin2-GFP together with mRFP-CLC have shown that peaks of dynamin 2-GFP recruitment correspond to an average recruitment of about 35 dynamin molecules per CCP[35] (Fig. 1i–k). We confirmed that a similar number of dynamin 2 molecules is recruited to CCPs upon plasmid-based re-expression of siRNA-resistant dynamin 2-GFP[43] in genome-engineered cells depleted of endogenous dynamin 2 (Fig. 1l), validating our approach. Expression of GFP-PI3KC2α in SK-MEL-2 cells depleted of endogenous PI3KC2α revealed an average recruitment of about 33 PI3KC2α molecules per CCP. When depleted of endogenous SNX9 and its close paralogue SNX18 and

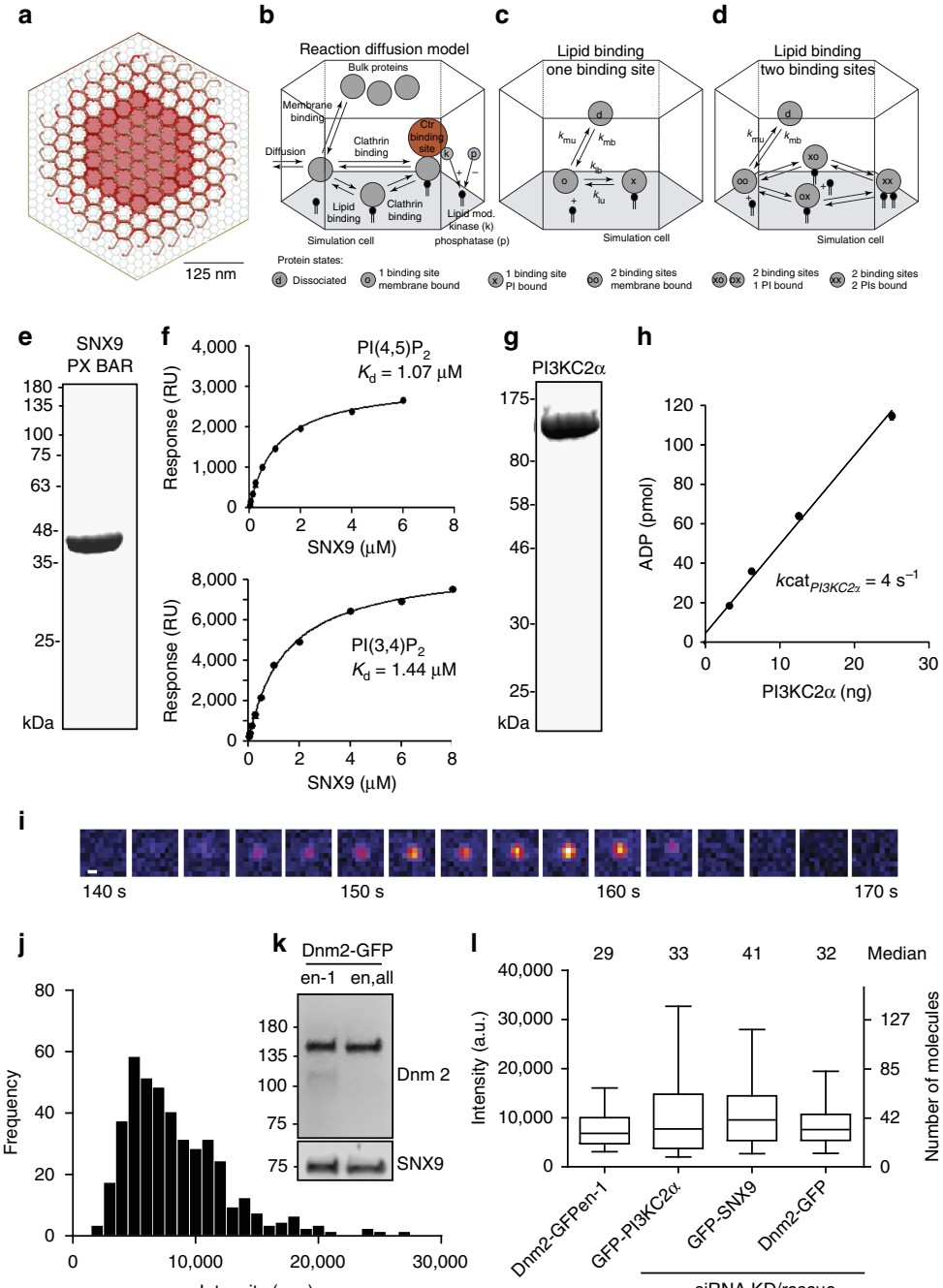

**Figure 1 | Reaction-diffusion model and experimental determination of model parameters.** (**a**) Reaction-diffusion master-equation model of a CCP surface using a hexagonal discrete lattice and 180 clathrin triskelia. Red: area eventually endocytosed corresponding to an ~150 nm diameter vesicle. (**b**) Summary of the reaction and binding processes occurring in simulation cells. (**c,d**) Sketch of binding model for proteins with one/two PI-binding sites, respectively (see Methods and Supplementary Table 1 for further details). (**e**) Purity of recombinant SNX9 PX-BAR domain (SNX9[204-595]). (**f**) Binding of the SNX9 PX-BAR domain to phosphoinositide-containing membranes (68% PC, 30 % PE, 2% PI(3,4)$P_2$ or PI(4,5)$P_2$) on L1 biosensor (reference channel: 70% PC/30 % PE). Sensorgram (Supplementary Fig. 1) fitting resulted in the $K_D$s. Of note, the exact $K_D$ value may be affected by the association of the BAR domain with additional (for example, non-PI) lipids, which, however, is insufficient to stably recruit SNX9 to membranes (compare Fig. 3). (**g**) Purity of recombinant N-terminally truncated PI3KC2α. (**h**) PI3KC2α kinase titration curve, performed at 37 °C for 15 min with 100 μM ATP and 400 μM PI(4)P. PI3KC2α specific activity: 1,803.6 nmol min$^{-1}$ mg$^{-1}$, derived $k_{cat}$: 4.0 s$^{-1}$. (**i**) TIRF microscopy recruitment timing of dynamin 2 (Dnm2)-GFP in SKMel cells. Scale bar, 200 nm. (**j**) Histogram of maximum fluorescence intensities minus background (mean of median pixels in 10 frames before and after maximum) of Dnm2-GFP from SKMel2 Dnm2-GFP,en-1 ($n=$ 420 CCPs from 42 cells in three experiments). Note that the first peak of the histogram represents one helix turn of Dnm2 of 26 molecules (similar to ref. 35). (**k**) Immunoblot analysis of SKMel2 cells with one (Dnm2-GFP,en-1) or both (Dnm2-GFPen,all) DNM2 alleles endogenously tagged with GFP. Untagged Dnm2 accounts for only ~10% of the Dnm2 signal (lower band). SNX9 serves as loading control. (**l**) Quantification of endogenous Dnm2 or PI3KC2α, SNX9 and Dnm2 after KD/rescue in SkMel2 CLC-mRFPen-all (three experiments each: Dnm2-GFPen-1, 420 CCPs, 42 cells; GFP-PI3KC2α, 290 CCPs, 29 cells; GFP-SNX9, 310 CCPs, 31 cells; Dnm2-GFP, 240 CCPs, 24 cells). Boxplots: median, 25th and 75th percentile, whiskers: 5th and 95th percentile.

functionally rescued by re-expression of GFP-SNX9, we found an average peak recruitment of 41 SNX9 molecules per CCP (Fig. 1l).

Our computational model uses an experimentally derived concentration of freely available $PI(4,5)P_2$ (13,750 $PI(4,5)P_2$ per µm (refs 2,44), additional references in Supplementary Information), and zero $PI(3,4)P_2$ (Supplementary Table 1) prior to PI3KC2α arrival at CCPs, corresponding to the most conservative assumption. Given fixed PI concentrations and cytosolic protein concentrations, the simulation will settle in a steady state, in which the copy numbers of endocytic proteins at the CCP fluctuate around a mean. The cytosolic protein concentrations were set such that this steady state achieves the reference protein numbers on average, corresponding to a strong population of $PI(4,5)P_2$ binders such as AP-2, whereas SNX9 was set to 20 molecules initially, corresponding to half its peak value (Fig. 2a, timescale 0–10 s).

As not all modelling parameters were unambiguously set by experimental data, the robustness of our model was additionally validated by testing 162 different parameter combinations, resulting in consistent findings (Supplementary Table 2).

**Role of PIP$_2$ phospholipids in the recruitment of SNX9.** The time course of PI3KC2α recruitment measured by live cell total internal reflection fluorescence (TIRF) microscopy[29] implies that $PI(3,4)P_2$ is produced during CCP maturation. We determined the $PI(3,4)P_2$ production rate by expressing catalytically active PI3KC2α recombinantly in insect cells and purifying the enzyme to homogeneity (Fig. 1g). Purified PI3KC2α synthesized $PI(3,4)P_2$ with a $k_{cat}$ of $4 \, s^{-1}$ ($V_{max} = 1,803.6 \, nmol \, min^{-1} \, mg^{-1}$) (Fig. 1h). These data allowed us to computationally model $PI(3,4)P_2$ synthesis at endocytic sites and its effect on protein populations. The $PI(4,5)P_2$ concentration was kept constant as PI(4)P 5-kinases are absent from CCPs[45].

Unrestricted lipid diffusion in a membrane bilayer is fast[46] and combined with the determined production rates for $PI(3,4)P_2$ and PI3KC2α copy numbers leads to rapid loss of $PI(3,4)P_2$. In this scenario, $PI(3,4)P_2$ production would have no effect on endocytic protein populations (Fig. 2a). Recent data, however, suggest that early-acting endocytic proteins result in the formation of stable lipid microdomains as a consequence of lowered lateral lipid diffusion by at least two, possibly three orders of magnitude[47,48]. The presence of a dense clathrin coat will likely restrict diffusion even further. Our simulations demonstrate that with a 200-fold slowdown of lipid diffusion underneath the assembled endocytic protein coat, the activity of 33 PI3KC2α kinases is sufficient to sustain a local pool of $PI(3,4)P_2$ with a concentration roughly half of that of $PI(4,5)P_2$ (Fig. 2b). In control simulations, using 500-fold diffusion slowdown, $PI(3,4)P_2$ rises to about 3/4 of the concentration of $PI(4,5)P_2$ (Fig. 2c). These results suggest that PI3KC2α recruited to nascent CCPs can indeed generate a stable local pool of $PI(3,4)P_2$, if lipid diffusion is restricted by endocytic protein coats. In all subsequent simulations, we used the more conservative 200-fold diffusion slowdown. Note that our protein recruitment simulations are conducted on a flat membrane, which also corresponds to a conservative setting. Endocytic intermediates are U or Ω-shaped, which involves an additional diffusion bottleneck around the clathrin coat that reduces the flux of produced $PI(3,4)P_2$ outside the CCP, and thus enhances the effects described here.

How does such local enrichment of $PI(3,4)P_2$ then affect endocytic protein copy numbers? Computational modelling of endocytic protein copy numbers under these conditions revealed a prominent enrichment of SNX9 at CCPs concomitant with rising levels of $PI(3,4)P_2$ (Fig. 2b). The copy numbers of other endocytic PI-binding proteins also respond, but experience very little relative change (Fig. 2b). Strong $PI(3,4)P_2$-dependent SNX9 recruitment was observed under a wide variety of parameter combinations (Fig. 2c,d,f; Supplementary Table 2) in spite of the near-equal affinity of SNX9 for $PI(3,4)P_2$ and $PI(4,5)P_2$ (Fig. 1f; Supplementary Table 1b). These results from computational modelling identify SNX9 as a candidate endocytic $PI(3,4)P_2$ effector protein that may aid the timed constriction of U-shaped endocytic intermediates.

As the lipid-binding preference of SNX9 could not explain its selective response to $PI(3,4)P_2$, we conducted control simulations to pinpoint, which properties of the protein–lipid interaction network are required for this behaviour, and, thus, render SNX9 a facilitator of endocytic membrane constriction. As successful endocytosis events were measured to involve an average peak recruitment of 41 SNX9 molecules per CCP *in vivo* (Fig. 1l), we regard 40 SNX9 molecules as sufficient to drive constriction. We consider a control simulation to describe a successful $PI(3,4)P_2$-driven SNX9 recruitment, if SNX9 starts below 30 copies and reaches at least 40 copies (shaded area in Fig. 2).

Eliminating binding to clathrin did not impair SNX9 recruitment to the membrane in the simulation, although it conceivably might affect the localization or the membrane-deforming activity of recruited SNX9 (Fig. 2d). If SNX9 was rendered fully $PI(3,4)P_2$-specific by artificially removing its affinity for $PI(4,5)P_2$, a selective response of SNX9 was maintained, but the final steady-state number of recruited SNX9 molecules stayed well below the number required for successful constriction (Fig. 2e). If, on the other hand, competition by other endocytic PI-binding proteins was eliminated, the initial copy number of SNX9 molecules was already well above the constriction threshold, thereby disabling the role of $PI(3,4)P_2$/SNX9 as a constriction switch (Fig. 2f).

These results suggest that selective recruitment of SNX9 as a $PI(3,4)P_2$-triggered switch for timed membrane constriction results from its ability to bind to either $PI(3,4)P_2$ or $PI(4,5)P_2$. This promiscuity provides SNX9 with a selective advantage over competing PI-binding endocytic proteins once $PI(3,4)P_2$ synthesis commences.

We tested the prediction that membrane recruitment of SNX9 requires its association with $PI(3,4)P_2$ or $PI(4,5)P_2$ experimentally. We capitalized on the observation that SNX9 along with other BAR domain proteins enriches at late-stage CCPs accumulating in cells lacking[23] or depleted of dynamin 2 (ref. 29) (Fig. 3). Eliminating the ability of SNX9 to specifically associate with $PI(3,4)P_2$ or $PI(4,5)P_2$ via the same site on its PX domain rendered the protein completely soluble, while a mutant lacking its dynamin-binding SH3 domain was recruited normally to stalled CCPs in dynamin 2-depleted cells (Fig. 3). Mutational inactivation of the interface for negatively charged lipids on the BAR domain or the amphipathic helix of the PX-BAR module also resulted in largely soluble SNX9 with little CCP enrichment. Collectively, these theoretical and experimental data provide a rational explanation for the specific recruitment of SNX9 by $PI(3,4)P_2$ despite the lack of selectivity of SNX9 for $PI(3,4)P_2$ vs. $PI(4,5)P_2$ observed in biochemical assays[29,41] (Fig. 1). They suggest a mechanism whereby available $PI(4,5)P_2$ maintains basal levels of SNX9 at CCPs that are selectively elevated by $PI(3,4)P_2$ production within a nonlinear interaction network of PI-competing endocytic proteins to drive progression of the endocytic reaction towards membrane fission.

**Timed recruitment of SNX9 to the endocytic vesicle neck.** Previous data by us and others[29,49] suggest that SNX9 is recruited to CCPs about 20 s after the $PI(3,4)P_2$-synthesizing enzyme PI3KC2α synchronous to the transition from U-shaped to

Ω-shaped CCP, but prior to the recruitment of dynamin[29]. However, as one earlier study had suggested a very late recruitment of SNX9 to CCPs post-dynamin-mediated fission[50]

we first reinvestigated the time course of SNX9 recruitment using genome-engineered SK-MEL-2 cells co-expressing mRFP-CLC together with dynamin 2-eGFP from their endogenous loci[35].

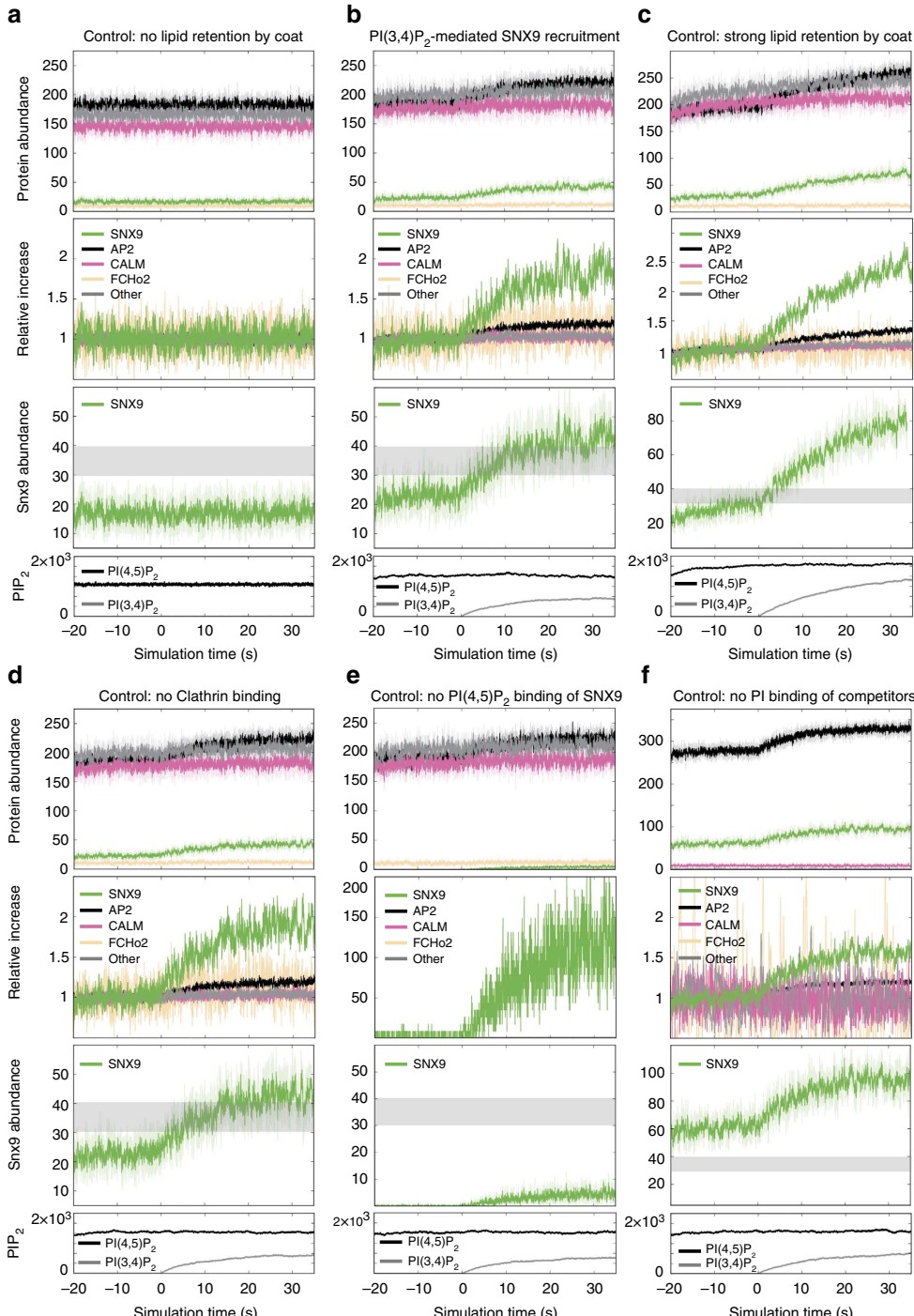

**Figure 2 | PI3KC2α-mediated PI(3,4)P₂ synthesis is predicted to selectively facilitate recruitment of SNX9.** Simulated endocytic protein numbers underneath coat of a 2D-projected CCP over time. Top rows: averaged protein copy numbers (solid lines) ± s.d. from four simulation runs (shaded regions). Second from top rows: ratio of protein copy numbers normalized to those at time $t = 0$ s. Second from bottom rows: SNX9 copy number. Grey area indicates the range of SNX9 that must be crossed to be considered a PI(3,4)P₂-triggered SNX9 recruitment that can drive an endocytosis event. Bottom rows: concentration profile of PI(3,4)P₂ and PI(4,5)P₂ during initial phase ( − 20 to 0 s), and PI(3,4)P₂ production phase (0–35 s). (**a**) Control simulation without lipid diffusion retention under the coat. Lipids diffuse away fast and PI(3,4)P₂ production does not lead to a local lipid pool. (**b**) Lipid diffusivity is decreased 200-fold under the coat. PI(3,4)P₂ accumulates and leads to a specific recruitment of SNX9 acting as a constriction switch to reach 42 SNX9 copies. (**c**) Lipid diffusivity is decreased 500-fold under the coat. (**d**) Control simulation where protein-clathrin affinity was set to zero. (**e**) Control simulation in which PI(4,5)P₂ affinity to SNX9 was set to zero. (**f**) Control simulation where the competition is lost by inactivating PI binding of other endocytic proteins, resulting in above-threshold recruitment of SNX9 at all times.

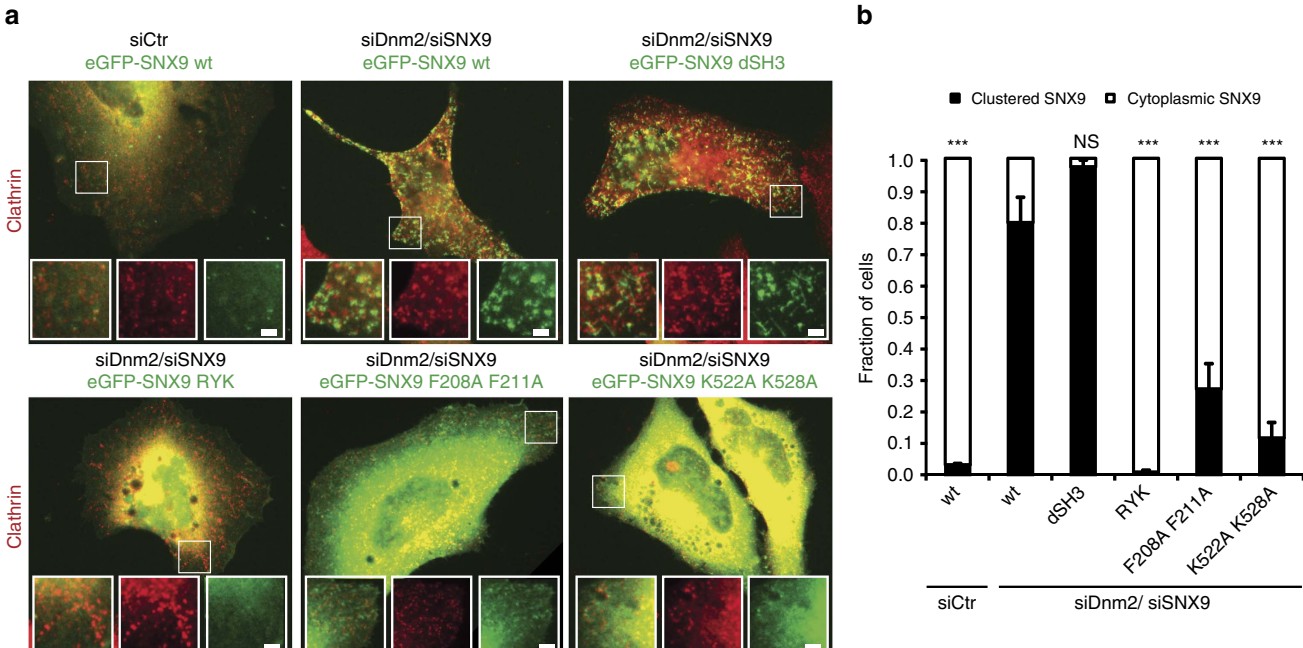

**Figure 3 | SNX9 recruitment to late stage CCPs requires its association with PI(3,4)P$_2$/PI(4,5)P$_2$.** (a) Representative epifluorescence images illustrating the localization of wild-type (wt) or mutant eGFP-SNX9 in control cells or cells depleted of endogenous dynamin 2 and SNX9 (siDnm2/siSNX9). RYK, PI-binding-defective PX domain mutant[29,62]; F208A,F211A, amphipathic helix mutant; K522A,K528A, membrane binding-defective BAR domain mutant[62]; dSH3, truncation mutant lacking SH3 domain. Scale bar, 2 μm. (b) Quantitative analysis of eGFP-SNX9 wt and mutant localization as illustrated in a. Data represent mean ± s.e.m. (194, 187, 141, 168, 138 and 136 cells from three independent experiments were analysed, *** indicate $P < 0.001$ from one-way ANOVA with Dunnett's Multiple Comparison Test with siDnm2/ siSNX9 SNX9 wt set as reference).

We followed CCP dynamics in SK-MEL-2 cells depleted of endogenous SNX9 and its close paralogue SNX18 functionally rescued by re-expression of iRFP-SNX9. We found iRFP-SNX9 to be recruited late during the endocytic process, for example, tens of seconds after clathrin, but prior to dynamin (Fig. 4a,b). These results confirm previous data by us and others[29,49] that SNX9 localizes transiently to late-stage CCPs prior to dynamin when the transition from U-shaped to Ω-shaped CCPs[29] occurs.

Based on its time course of recruitment and its binding to PI(3,4)P$_2$/PI(4,5)P$_2$, clathrin, AP-2 and dynamin, newly recruited SNX9 could conceivably localize either to the clathrin-coated dome or to the neck of maturing CCPs. To investigate this we explored the nanoscale spatiotemporal distribution of SNX9 at CCPs by high-resolution interacting-particle reaction-diffusion (iPRD) simulations[51,52], generating predictions that were then tested experimentally by super-resolution imaging. The ReaDDy software[51,53] was used to model U- and Ω-shaped CCPs fitting idealized shapes to specific CCP intermediates observed by EM (Fig. 4c,h). Endocytic proteins were positioned on the CCP with numbers, spatial localizations and clathrin binding site occupations that corresponded to late representative snapshots at steady-state PI(3,4)P$_2$ of the reaction-diffusion simulations described in Fig. 2b. ReaDDy resolves the spatial trajectories and the excluded space due to physical size of each protein copy. The dense protein matrix underneath the clathrin lattice (Fig. 4c,h) resulted in space exclusion and, thus, in a predominant recruitment of SNX9 to the CCP rim (Fig. 4d,e). Brownian dynamics simulations further demonstrated that the preferential localization of SNX9 at the neck persists over time (Fig. 4f). Since SNX9 has been shown to oligomerize when bound to membranes[41], and other BAR-domain proteins have been demonstrated to form filaments and lattices involving tip-to-tip interactions[14], we also tested the effect of homophilic interactions between SNX9 tips. Modest binding energies of 5 to 10 k$_B$T,

corresponding to the formation of one to two salt bridges[54], were sufficient for the formation of linear SNX9 assemblies (Supplementary Fig. 2c). At narrow CCP necks, these linear assemblies form a closed ring (Fig. 4g). SNX9 molecules initially present under the clathrin cage could diffuse to the neck and were then trapped there by oligomer interactions with these ring-like assemblies (Fig. 4i). Ring fragments assembled to wide necks may contribute to constriction as BAR domains can spontaneously induce local membrane curvature[55–57] and may, thus, be able to reduce the neck diameter successively during recruitment (Fig. 4g).

Our data suggest that PI(3,4)P$_2$ together with weak homophilic interactions between recruited SNX9 molecules may drive formation of SNX9 rings or even lattices around CCP necks. Such BAR oligomer assemblies will affect the local membrane curvature and may drive the progression to the constricted state, as demonstrated previously *in silico*[55–57]. Simulations of the constricted state of CCPs predicted SNX9 to form a ring at the neck (Fig. 4g) that upon projection into the *xy* plane (Fig. 4j) would exhibit a single ring of about 100 nm in diameter when viewed at a lateral resolution of 25 nm (as achieved in super-resolution imaging, see below).

To test these predictions experimentally, we employed direct stochastic optical reconstruction-based super-resolution (*d*STORM) microscopy, an imaging technique that achieves a lateral resolution of about 25 nm. To accomplish nanoscale resolution of the endocytic protein distribution, we stalled CME at the level of constricted Ω-shaped endocytic intermediates in cells by expressing mutant dynamin 2 (K44A). This allowed the centred alignment of hundreds of endocytic CCPs to create average fluorescent intensity maps that when projected two-dimensionally provide a high-resolution picture of endocytic protein organization for both endogenous as well as eGFP-tagged proteins (Fig. 5a,b; Supplementary Fig. 3). Clathrin exhibited a ring-like distribution reflecting a two-dimensional projection of

the cage organization of stalled CCPs with a mean diameter of about 130–140 nm, consistent with the outer diameter of invaginated Ω-shaped CCPs measured by electron microscopy ($161 \pm 20$ nm ($n = 30$)). A very similar distribution pattern was observed for the PI(3,4)P$_2$-synthesizing enzyme PI3KC2α, which directly associates with clathrin[58] (Fig. 5c). As PI3KC2α is activated by clathrin, PI(3,4)P$_2$ synthesis is likely to occur under the entire clathrin lattice. This organization would maximize access of the enzyme to its substrate PI(4)P on the plasma

membrane[59]. Native SNX9 or its eGFP-tagged variant also displayed a ring-like pattern; however, these rings were significantly narrower than those observed for clathrin and PI3KC2α with a mean diameter of about 100 nm, but slightly wider than those seen for dynamin 2 itself (Fig. 5b,c), consistent with our simulations (Fig. 4j). These data from 2D-*d*STORM support a model in which SNX9 assembles into ring-like structures around the endocytic vesicle neck. To directly test this model we determined the z-position of SNX9 and PI3KC2α

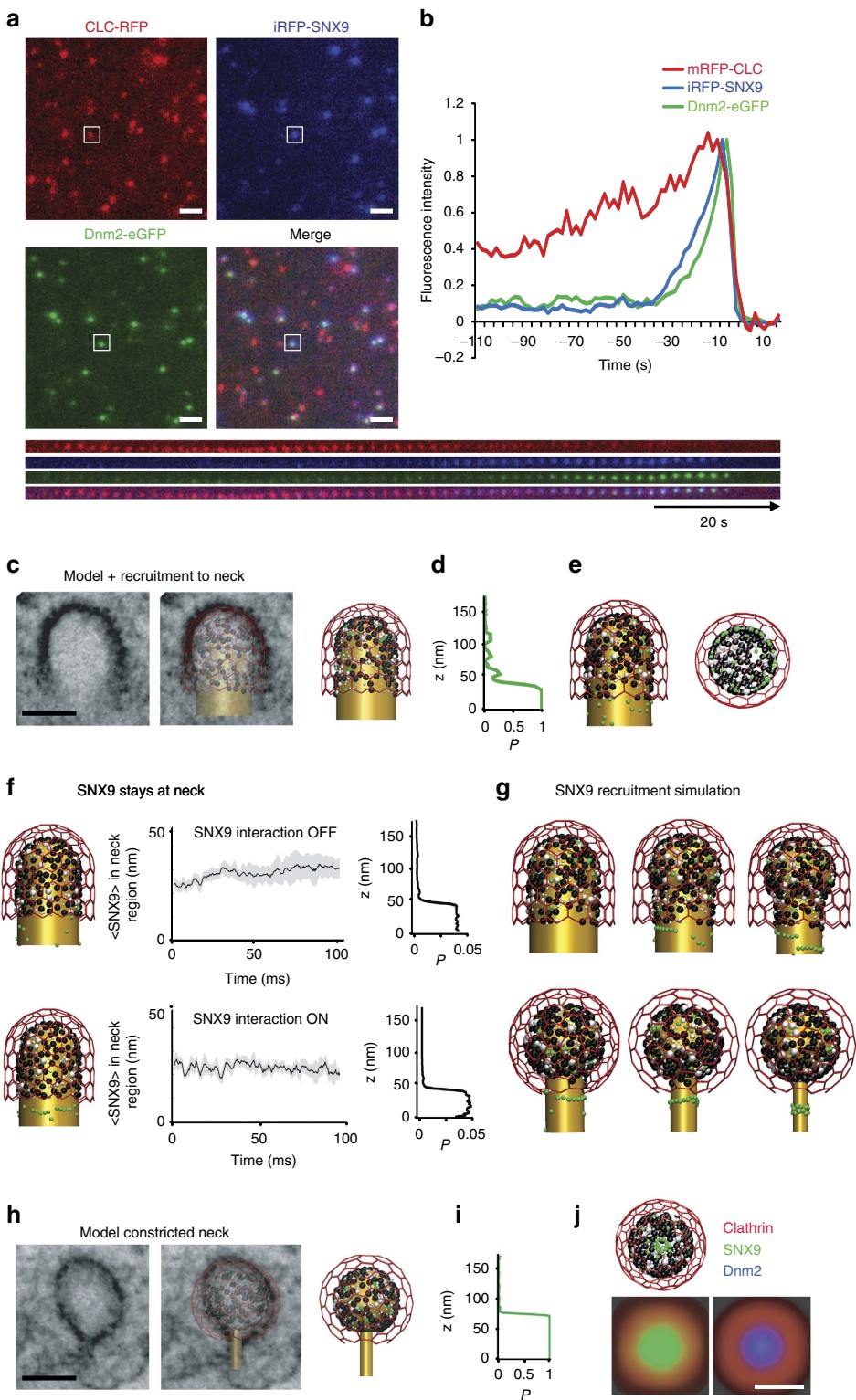

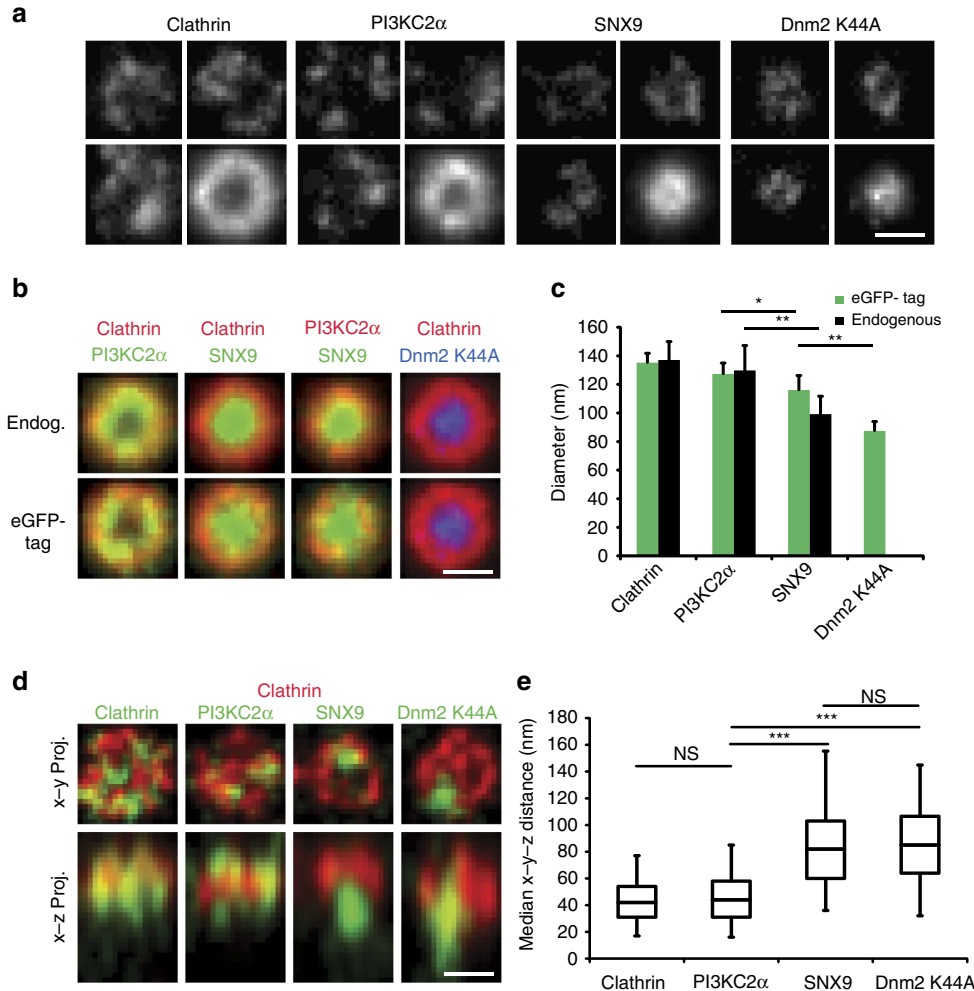

**Figure 5 | SNX9 nanoscale localization to the neck of late stage CCPs.** (**a**) dSTORM images of endogenous clathrin, PI3KC2α and SNX9 or Dnm2-eGFP at late stage Ω-CCP in Cos7 cells expressing GTPase-deficient dynamin 2 (Dnm2) K44A-eGFP. Centre-aligned average images (bottom right quarters) were generated from 180–762 CCP from three to nine cells. (**b**) Overlay of centre-aligned averaged dSTORM images. (**c**) Quantification of diameter of averaged structures from endogenous and eGFP-tagged proteins. Data represent mean ± s.d. ($n = 180$–763 CCPs from three to nine cells) (**d**) 3D-dSTORM images of eGFP-tagged clathrin, PI3KC2α, SNX9 and Dnm2 K44A together with endogenous clathrin in Dnm2-K44A expressing Cos. Shown are representative $x$–$y$ projections and $x$–$z$ projections from $200 \times 200 \times 1,200$ nm volumes centred around individual CCPs (**e**) Quantification of the distance of the median from clathrin towards the indicated markers (Clathrin: 647 CCPs from nine cells from three experiments, PI3KC2α: 746 CCP from 10 cells from three experiments, SNX9: 725 CCP from 12 cells from five experiments, Dnm2-K44A: 488 CCPs from 13cells from five experiments). Statistical test on **c,e** was performed with Student's $t$-test; NS, non-significant, ***$P < 0.001$, **$P < 0.01$ and *$P < 0.05$, Scale bars, 100 nm.

**Figure 4 | Spatiotemporal model of SNX9 recruitment and nanoscale localization.** (**a**) Representative TIRF images illustrating the localization of iRFP-SNX9, endogenously tagged clathrin light chain (CLC)-mRFP and dynamin 2 (Dnm2)-eGFP in SK-MEL-2 cells depleted of endogenous SNX9/18. Scale bar, 2 μm. (**b**) Average recruitment of iRFP-SNX9 and endogenously tagged CLC-mRFP and Dnm2-eGFP in SK-MEL-2 cells ($n = 60$ CCPs from four cells). (**c**–**e**) High protein densities under the clathrin coat are predicted to lead to SNX9 recruitment predominantly towards the neck. (**c**) Comparison of computational model and corresponding EM image in a pre-constricted state. The membrane (gold) forms a U-shaped structure coated by AP-2 (black), other endocytic proteins (grey) and clathrin (red). Few SNX9 molecules (green) are present in the coat due to PI(4,5)P$_2$-binding. Scale bar, 100 nm. (**d**) Probability to recruit SNX9 to a given position on the $z$ axis. Space exclusion in the protein-dense dome region leads to most SNX9s being recruited to the neck. (**e**) Configuration after 20 SNX9 molecules have been recruited (side and bottom view). (**f**) Brownian dynamics simulations show that SNX9 remains predominantly in the neck region (three independent simulations, grey depicts s.d.). This is the case when both (top) SNX9 molecules do not interact, and when (bottom) SNX9 forms oligomers by a tip-to-tip interaction potential promoting ring formation (see Supplementary Information and Supplementary Fig. 2). Left: snapshot of the simulation, middle: time evolution of the SNX9 fraction at the neck, right: SNX9 $z$-coordinate residence probability. (**g**) Snapshots of neck constriction and Snx9 recruitment simulations. SNX9 tip-to-tip interactions induce oligomer formation. (**h**) Constricted neck simulation model is based on EM imaging (compare Fig. 2, colour as in **c**). The higher curvature increases the protein density under the coat. Scale bar, 100 nm. (**i**) Probability to find SNX9 molecules at a given $z$ position. SNX9 is almost completely located at the neck, as the increased coat protein density allows for few vacancies. (**j**) Structure shown from the bottom after SNX9 rings have formed and as a simulated STORM image from the bottom view using a Gaussian blurring filter that mimics a 25 nm resolution (compare to **g,h**).

with respect to endogenous clathrin and dynamin 2 (K44A) using dual-colour 3D-*d*STORM super-resolution imaging (Fig. 5d). Quantification of the median distance of SNX9 towards clathrin revealed its concentration at endocytic vesicle necks, where it overlapped with dynamin 2, while PI3KC2α localized to the clathrin coat (Fig. 5d,e).

These results suggest that timed lipid-driven recruitment of SNX9 causes its localization and concentration at the endocytic vesicle neck, possibly at the interface with the clathrin coat. While PI(3,4)P$_2$ is produced under the protein-dense clathrin coat, SNX9 assembles at the neck where it finds sufficient space, oligomeric contacts to engage in, and, presumably, also a preferred membrane curvature. Consistently, the SNX9 concentration is not found to be increased at the flat membrane around the CCP, although this area is expected to contain elevated levels of PI(3,4)P$_2$.

**SNX9 couples CCP constriction to dynamin-mediated fission.** Overall, the data described thus far suggest a mechanism, in which PI(3,4)P$_2$ distributed over the entire endocytic intermediate recruits SNX9 to the CCP neck in a selective and localized manner. We hypothesize that its assembly couples membrane constriction to dynamin assembly and fission. In order to test this hypothesis experimentally, we employed a knockdown/rescue approach. Cells depleted of endogenous SNX9 in combination with its close paralogue SNX18 (see also ref. 29) (Fig. 6a) displayed grossly attenuated CCP dynamics that were rescued by expression of siRNA-resistant wild-type SNX9 (Fig. 6b,c). If SNX9 indeed operates at the neck of CCPs to aid membrane constriction prior to dynamin-mediated fission, one would expect that cells depleted of SNX9/SNX18 accumulate CCPs with wide, open necks akin to cells lacking PI3KC2α (ref. 29). SNX9/SNX18-deficient Cos7 cells indeed revealed increased numbers of U-shaped CCPs, whereas the frequencies of early shallow CCPs, Ω-shaped constricted CCPs, or of free CCVs were unaltered (Fig. 6d,e). An accumulation of U-shaped intermediates was also overt from *d*STORM imaging of the distribution of dynamin 2 (K44A): the mean diameter of dynamin 2 rings in SNX9/18-depleted cells was significantly wider than in control cells (Fig. 6f), while the overall levels of dynamin 2 recruited to CCPs were slightly increased in absence of SNX9/18 (compare Fig. 7d and below), suggesting that SNX9/18

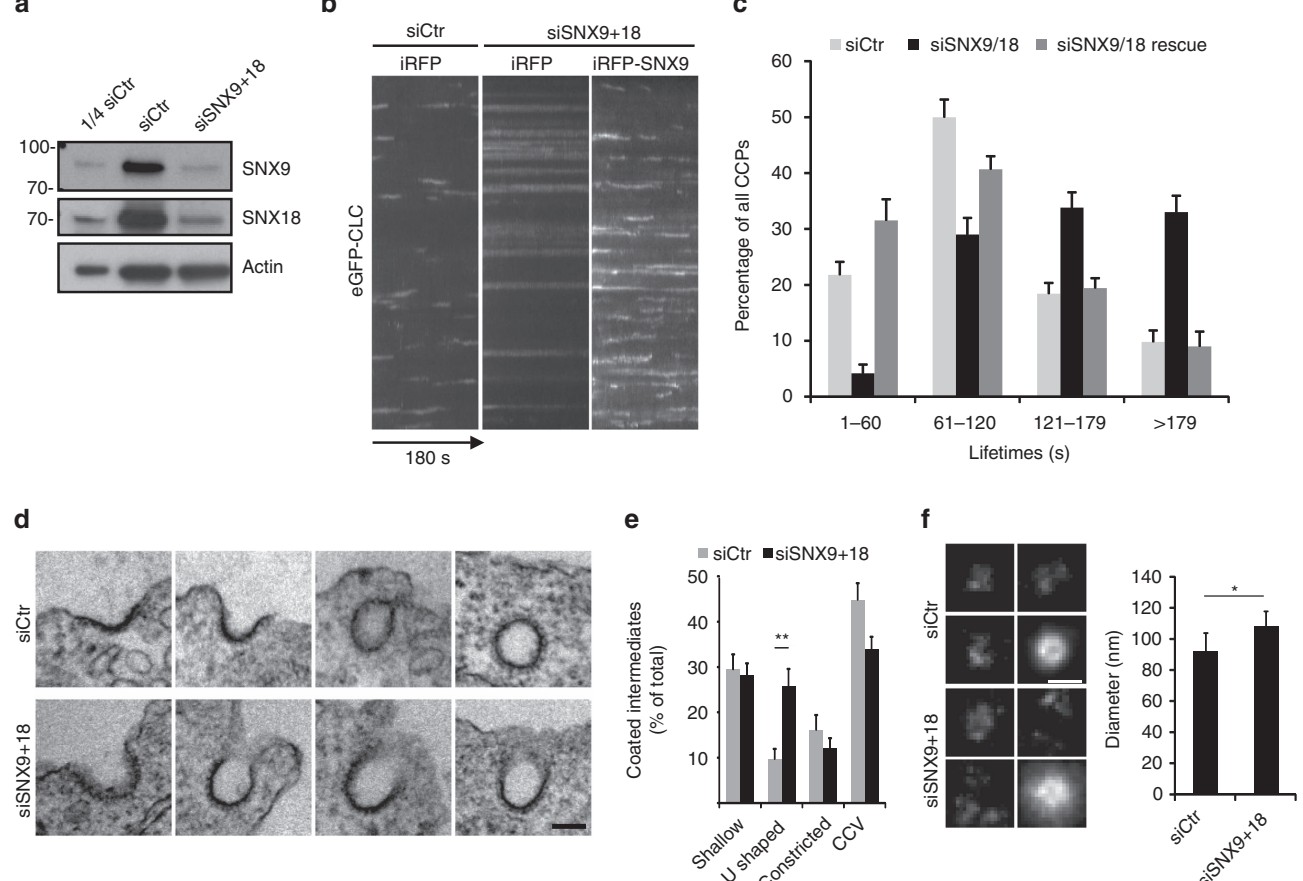

**Figure 6 | SNX9 is required for late-stage maturation and constriction of CCPs.** (**a**) Efficient siRNA-mediated depletion of SNX9 and SNX18 in Cos7 cells. (**b,c**) SNX9/18 depletion impairs CCP dynamics analysed by TIRF imaging of eGFP-clathrin expressing Cos7 cells depleted of SNX9/18. (**b**) Kymographs show increased CCP-lifetimes in cells depleted of SNX9/18. (**c**) Lifetime distribution of CCPs binned in categories of 60 s. Data represent mean ± s.e.m. (*n* = 10 cells from two independent experiments and a total of > 500 CCPs analysed per condition. (**d,e**) Ultrastructural analysis of CCPs in control or SNX9/18-depleted cells. Morphological groups were shallow, non-constricted U-shaped, constricted Ω-shaped pits, or structures containing complete clathrin coats. (**d**) Representative images from controls (top) or a SNX9/18-depleted cell illustrating accumulation of U-shaped pits (bottom). Scale bar, 100 nm. (**e**) Bar diagram detailing the relative abundance of different clathrin-coated structures in **d** control versus SNX9/18-depleted Cos7 cells (mean ± s.e.m.; *n* = 19–23 cell perimeters, two-sided Student's *t*-test, **P < 0.01). (**f**) *d*STORM and centre-aligned average images (bottom right quarters) and average diameter of Dnm2-eGFP K44A in control or SNX9/18-depleted cells (mean ± s.d., *n* = 7 cells, 1,370–1,440 CCPs, two-sided Student's *t*-test, *P < 0.05).

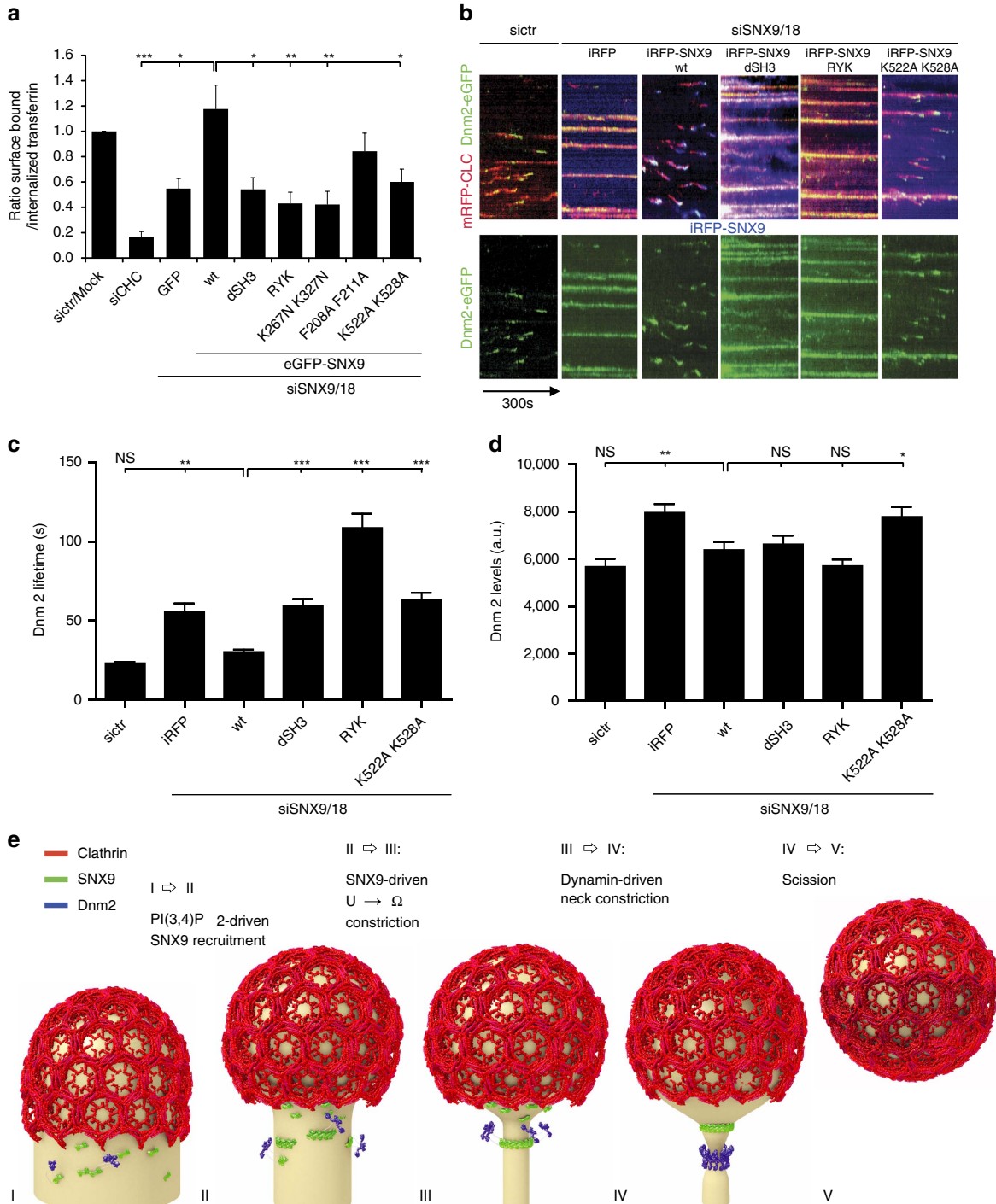

**Figure 7 | SNX9 couples constriction of late-stage CCPs to dynamin-mediated fission in CME.** (**a**) Impaired CME of transferrin in Cos7 cells depleted of endogenous SNX9 and its close paralogue SNX18 is rescued by re-expression of wild-type (wt) eGFP–SNX9 but not of SNX9 mutants lacking the SH3 domain (dSH3), the ability to bind to PI(3,4)P$_2$/PI(4,5)P$_2$ via their PX domain (RYK and K267N, K327N; refs 29,41), or carrying mutations within the amphipathic helix (F208A, F211A), or the BAR domain (K522A, K528A)[62]. Bar diagrams represent the ratio of internalized (10 min, 37 °C) to surface transferrin (45 min, 4 °C) (mean ± s.e.m; $n = 3$–5 experiments). (**b**) Representative kymographs of CCP-lifetimes in genome-edited SK-MEL-2 cells expressing clathrin light chain-mRFP (mRFP-CLC) and dynamin 2-eGFP (Dnm2-eGFP) from endogenous loci. Cells were depleted of endogenous SNX9/18 followed by re-expression of iRFP-SNX9 wt or mutant proteins as indicated. (**c**) Dnm2-eGFP lifetimes and (**d**) levels of Dnm2-eGFP of randomly chosen CCPs (mean ± s.e.m., $n = 120$ CCP cells from eight cells from two independent experiments) Statistical test for **a**,**c**,**d** was performed with one-way ANOVA with Dunnett's Multiple Comparison Test with siSNX9/18 SNX9 wt set as reference ***$P < 0.001$, **$P < 0.01$ and *$P < 0.05$. (**e**) (I) U-shaped CCP with clathrin coat (red), endocytic proteins under the coat (not depicted). SNX9 molecules (green), partially associated with dynamin (blue) are bound to PI(3,4)P$_2$. (I→II): Clathrin-associated PI3KC2α catalyses PI(4)P→PI(3,4)P$_2$ conversion at CCPs leading to selective recruitment of SNX9 from the cytosol to the CCP neck. (II) SNX9 forms oligomers that (II→III) enwrap the CCP neck as rings or spirals and bend the membrane, driving constriction from U-shaped to Ω-shaped endocytic intermediates. (III→IV) Dynamin associates with the narrow neck and oligomerizes to drive further constriction. (IV→V) Superconstricted membrane neck breaks as a result of neck constriction[83,84]. (V) Release of endocytic vesicle.

are dispensable for recruitment of dynamin 2 to endocytic pits. Defective CME due to accumulation of U-shaped CCPs was also seen in SNX9-depleted HeLa cells (which express very low levels of SNX18; Supplementary Fig. 4), while depletion of dynamin 2 resulted in the accumulation of constricted, often elongated necks of CCPs (Supplementary Fig. 4d,e), consistent with earlier data[23]. These results demonstrate that SNX9/18 is required for constriction of CCPs, but is dispensable for enrichment of dynamin at endocytic sites, a function likely executed by other SH3 domain proteins such as amphiphysins[60,61]. Accumulation of U-shaped CCPs and stalled CCP dynamics have also been observed in PI3KC2α-depleted cells[29], consistent with recruitment of SNX9/18 by PI3KC2α-mediated PI(3,4)P$_2$ synthesis.

Finally, we asked which structural elements within SNX9 mediate membrane constriction and whether association with dynamin is required to couple constriction to the fission reaction. Wild-type and mutant versions of SNX9 were re-expressed in cells lacking endogenous SNX9/18 and their ability to rescue defective CCP dynamics and to support transferrin (Tf)-CME was analysed. SNX9 mutated within its PX domain (RYK or K267N, K327N) to abrogate specific binding to PI(3,4)P$_2$ or PI(4,5)P$_2$ resulted in soluble protein (Fig. 3b) that failed to rescue defective CME (Fig. 7a) and produced a dominant-negative effect on CCP dynamics in genome edited SNX9/18-depleted SK-MEL-2 cells co-expressing clathrin light chain (CLC)-mRFP and dynamin 2-eGFP at endogenous levels[35] (Fig. 7b). A similar though slightly less marked phenotype was observed, if the amphipathic helix (F208A, F211A) or the lipid-binding BAR domain (K522A, K528A) were mutated (Fig. 7a,b). Interestingly, a SNX9 mutant lacking its dynamin-binding SH3 domain was recruited to the plasma membrane (Fig. 3), but failed to restore Tf-CME or attenuated CCP dynamics (Fig. 7a,b; dSH3). Defective CME and stalled CCP dynamics in cells expressing mutant SNX9 were associated with significantly increased lifetimes of dynamin 2 at CCPs (Fig. 7c), while dynamin 2 levels at CCPs were either unaltered or slightly elevated compared to control cells or cells re-expressing WT SNX9 (Fig. 7d).

Collectively, these data indicate that SNX9/18 via its PX-BAR and SH3 domains couples constriction of endocytic CCPs to dynamin-mediated membrane fission of mature endocytic intermediates during CME.

## Discussion

Though a number of BAR domain proteins have been identified and proposed to act at distinct stages of endocytosis in yeast or mammalian cells it has remained unclear how their recruitment to endocytic sites is spatiotemporally controlled. Our data identify a lipid-based mechanism of BAR domain protein recruitment in which PI3KC2α-mediated synthesis of PI(3,4)P$_2$ within the clathrin-coated membrane area together with nonlinear interactions between endocytic proteins competing for a limiting number of lipid molecules mediate the local assembly of the PX-BAR domain protein SNX9 (and likewise SNX18) at the neck of invaginated coated pits (Figs 2 and 3). We provide multiple independent lines of evidence to support this hypothesis: data from computational modelling suggest that due to its time course of recruitment by PI3KC2α-mediated synthesis of PI(3,4)P$_2$, experimentally verified in genome-edited cells, SNX9 is recruited preferentially to the endocytic vesicle neck (Fig. 4). In line with this prediction we directly show by dual-colour 3D-dSTORM super-resolution imaging of the nanoscale organization of endocytic proteins at CCPs that SNX9 is concentrated at the neck of stalled endocytic intermediates (Fig. 5), for example, at a site distinct from the major site of PI(3,4)P$_2$ production by PI3KC2α within the clathrin coat, where substrate access is

maximal. Finally, combined experimental data from mammalian cells expressing fluorescent protein-tagged mutants of SNX9 as their sole variant indicate that SNX9 is required for the constriction of the endocytic vesicle neck and both its PX-BAR and SH3 domain modules are required to couple constriction to dynamin-mediated fission (Figs 6 and 7).

The molecular mechanism by which SNX9 promotes the constriction of the endocytic vesicle neck remains elusive. One possibility is that membrane constriction is driven by local re-arrangement of SNX9 molecules into their preferred assembly state enclosing a membrane stalk of 20–40 nm in diameter[62]. It is also possible that SNX9 assembly affects the tension of the underlying membrane, for example, by insertion of its amphipathic helix that we show to promote the function of SNX9 in CME (compare Fig. 7a–d), and this allows dynamin, perhaps in combination with additional components such as actin and actin regulatory factors associated with SNX9/18 and dynamin[35–37,62] and/or other BAR domain proteins (for example, amphiphysin, endophilin A1-3) to mediate constriction to form a 20–40 nm membrane stalk. Consistent with such a scenario, we observe a significant elevation of amphiphysin1 but not endophilin A2 levels in SNX9/18-depleted cells (Supplementary Fig. 5). Such a narrow stalk provides a membrane template ideally suited for assembly of dynamin in its non-constricted state[33] to eventually drive fission (Fig. 7e). Our data further show that dynamin 2 in spite of its tubulation activity on artificial lipid templates in vitro[13,21,31] is insufficient for efficient constriction of U-shaped endocytic intermediates in mammalian cells in vivo, consistent with recent results from model membrane templates[32]. PX-BAR protein-mediated constriction of invaginated CCPs is also consistent with recent correlative light and electron microscopic data showing that endocytic intermediates mature by continuous membrane remodelling[17] known to be driven by BAR domain proteins.

We suggest that timed recruitment of membrane remodelling BAR domain proteins is a general property of endocytic membrane internalization in all cells. Coupling BAR domain protein recruitment to the activity of a lipid kinase may provide mammalian cells with a regulatory mechanism that allows cells to adapt endocytic membrane deformation to physiological needs, for example, to enable selection of specific cargo prior to membrane constriction and fission. How such regulation occurs will need to be addressed in future studies.

## Methods

**Reaction-diffusion master equation model.** A reaction-diffusion master equation (RDME)[39,40,63] was developed in order to model the distribution and diffusion of PIs at the CCP site and recruitment of endocytic proteins to the membrane as response to PI concentration changes. See Fig. 1a-d for an illustration of the model. The simulation describes protein recruitment to a membrane area that is discretized using a two-dimensional hexagonal grid of simulation cells (Fig. 1a; Supplementary Table 1). The simulated area contains a prototypical CCP with a regular hexagonal coat of 180 clathrin molecules, giving rise to a lattice of 540 N-terminal binding domains, corresponding to an outer diameter of about 160 nm when forming a vesicle. The total simulation area is chosen ten times larger than the CCP to allow lipid diffusion out of the CCP to be modelled. Each clathrin triskelion exposes three binding sites to clathrin-binding endocytic proteins underneath, thus defining a spatial pattern of binding sites. These terminal binding domains are associated with specific simulation cells—such that a given simulation cell may have none, one or two terminal domains. At any time, each simulation cell maintains the copy numbers of PI(3,4)P$_2$ and PI(4,5)P$_2$ molecules, of membrane-associated copies of the proteins: SNX9, AP-2, FCHo2, CALM, other PIP binders, and the number of clathrin-binding sites, and which proteins are bound there. Each protein copy in the simulation maintains the number of PI(3,4)P$_2$ or PI(4,5)P$_2$ molecules bound.

The simulation proceeds in discrete time steps of length dt. For each protein or lipid, one reaction event of each type (see below) is considered per time step and is executed with the probability $P = 1 - \exp(-k\,dt)$, where $k$ is the rate. Reactions that involve insertion of a protein into a new cell (either by diffusion or

recruitment) are only executed if the target cell has sufficient space available (see Protein and lipid diffusion in the RDME model).

- Protein association/dissociation: endocytic proteins are recruited to the membrane with a rate constant $k_{on,membrane}$ from a fixed bulk concentration (Supplementary Table 1) and then added to the target simulation cell. Membrane-associated proteins that are bound to neither phosphoinositide nor clathrin may dissociate with a dissociation rate defined by the intrinsic microscopic binding affinity, $K_{a,mem}$ (Supplementary Table 1) and the binding rate. Membrane-associated proteins cannot dissociate while they bind at least one PI lipid or are bound to a clathrin-binding site.
- Lipid binding/unbinding: PI-binding sites of any membrane-associated protein can bind free phosphoinositides present in the same simulation cell with a rate $k_{on,lipid}$, or loose bound PI lipids with dissociation rates based on $k_{on,lipid}$ and the affinities $K_{a,lip45}$, $K_{a,lipweak45}$, $K_{a,lip34}$ and $K_{a,lipweak34}$ (Supplementary Table 1).
- Clathrin-protein association/dissociation: proteins can bind to free clathrin-binding sites in the same simulation cell with a rate $k_{on,clathrin}$ (Supplementary Table 1). Each clathrin-binding site can only associate with one protein at a time. The protein remains in the same simulation cell but is removed from the freely diffusing pool. Clathrin-associated proteins can dissociate with a rate according to $k_{on,clathrin}$ and the affinity $K_{a,clathrin}$ (Supplementary Table 1). The protein remains in the same simulation cell and is returned to the freely diffusing pool.
- Production of PI(3,4)P$_2$ by PI3KC2α: PI(3,4)P$_2$ is produced by phosphorylation from the implicit pool of PI(4)P. Since PI3KC2α is associated with clathrin terminal domains[29,58], the reaction is conducted at clathrin-binding sites with a rate $k_{kinase}$ (Supplementary Table 1). Every such reaction event increments the local PI(3,4)P$_2$ lipid count by one. The production of PI(3,4)P$_2$ is started after a simulation time of 20 s.
- Depletion of PI(4,5)P$_2$ by phosphatases: in some simulation parameter settings, we are considering the action of phosphatases that deplete the PI(4,5)P$_2$ pool with a rate of $k_{phosphatase}$ (Supplementary Table 1) at every clathrin-binding site.
- Protein and lipid diffusion: lipid molecules that are not bound to proteins and all membrane-associated proteins (together with their bound lipids) can diffuse laterally, modelled by jump events of lipids or proteins between neighbouring simulation cells, with jump rates computed from diffusion constants (see section Protein and lipid diffusion in the RDME model). We use absorbing boundary conditions for PI(3,4)P$_2$, that is, these lipids are removed when they reach the simulation boundary. To model local lipid diffusion slowdown effects of the CCP, grid cells in the core of the CCP can be set to slow down lipid diffusion compared to free membrane, a 200 × diffusion slowdown is an estimate from the recent literature[47,48].

**Protein and lipid diffusion in the RDME model.** For phospholipid diffusion, the experimental value of 3 μm$^2$ s$^{-1}$ (ref. 46) was employed for both PI(4,5)P$_2$ and PI(3,4)P$_2$, yielding a jump rate of 13482 s$^{(-1)}$ between neighbouring simulation cells. For proteins, it is considered that diffusion in the cytosol is much faster than lateral membrane diffusion. Thus, the lateral diffusion constant of membrane-associated proteins is determined by the diffusion of the bound PIs, here taken to be the lipid diffusion constant, divided by the number of binding sites of the protein. Whenever it was attempted to move a protein into a target cell (either through diffusion, or through protein recruitment from the cytosol), availability of sufficient space was tested. The probability with which such a move would be accepted was set to account for the actual space left in that particular cell. For example, if a cell can take four globular proteins of a certain size, and three are already residing inside the cell, the probability to successfully place a fourth one is very low because it is unlikely that the space at the randomly chosen insertion point is completely free of other particles causing overlaps. We have thus computed placement probabilities for proteins of different sizes, depending on the space already allocated on the target cell, and fitted this placement probability with the sigmoidal function $(1 + \exp(a\,r + b))^{-1}$ with $r$ being the fraction of the simulation cell area already occupied, and $a$ and $b$ the following fitting parameters (AP-2: $a = 18.8$, $b = -2.7$; SNX9 & FCHo2: $a = 13$, $b = -2.8$; CALM & Others: $a = 9.3$, $b = -3.1$).

**Parametrization of protein-lipid binding in the model.** We describe how experimentally measured lipid-binding affinities of endocytic proteins are transformed into their respective microscopic simulation parameters in the RDME model. The membrane dissociation constant of proteins is defined by:

$$K_D = \frac{[\text{Protein}][\text{Lipid}]}{[\text{Complex}]}$$

where [Protein] is the volume concentration of the solvated protein (for example, measured as moles per litre), [Complex] is the surface concentration of all proteins that are associated to the membrane (either stably or transiently) and [Lipid] is the surface concentration of lipids (for example, measured in particles per nm$^2$). $K_D$ is an experimentally measurable dissociation constant. When the lipid and complex concentrations are measured in the same unit, $K_D$ will have the same unit as the protein concentration (for example, moles per litre).

In order to convert the macroscopic entity $K_D$ into microscopic simulation parameters, we must specify the dynamical model that the simulation uses to represent protein binding and dissociation events. Consider Fig. 1b–d as an illustration of the binding dynamics. We consider a simulation cell which is a membrane patch with area $A_{cell}$. A hypothetical column of cytosol with height $h$ is considered, yielding a simulation cell volume $V_{cell}$ ($h$ and thus $V_{cell}$ can be chosen arbitrarily—they are just needed as intermediate quantities for the calculation). The volume concentration of protein is thus given by:

$$[\text{Protein}] = \frac{n_d}{V_{cell}}$$

where $n_d$ is the average copy number of solvated proteins considered in the simulation cell volume. We use the convention that all lipid and associated protein concentrations are measured in particle numbers per simulation cell area. Thus,

$$[\text{Lipid}] = n_{lipid}$$

is the number of lipids per surface area $A_{cell}$. As shown in Fig. 1c,d, proteins can associate from the cytosol to state O, in which no phospholipid is bound, and can dissociate from state O to the cytosol. The fact that, depending on the settings of the rates $k_{mb}$ and $k_{mu}$, proteins can be membrane-associated for some time even if no phospholipid is present represents the intrinsic membrane affinity of endocytic proteins. Usually, this intrinsic affinity is small and is controlled by the ratio $k_{mb}/k_{mu}$. From the associated state O, proteins may enter stably bound states (including at least one X), where the identifier X specifies which phospholipid-binding sites are filled. We use the convention that a protein cannot dissociate from the membrane while at least one of the binding sites is filled with a phospholipid. This is based on the reasoning that pulling a phospholipid out of the membrane would be energetically extremely unfavourable. The mechanism of dissociation in our model is thus that phospholipids must first clear their binding sites, each with rate $k_{lu}$, before being able to dissociate with rate $k_{mu}$. The concentration of membrane-associated proteins is the total number of proteins in any lipid binding states on the simulation area. For proteins with a single binding site, this is:

$$[\text{Complex}] = n_O + n_X$$

where $n_O$ counts membrane-associated proteins with empty binding sites and $n_X$ counts membrane-associated proteins with filled binding sites. In proteins with two binding sites, we have the configurations:

$$[\text{Complex}] = n_{OO} + n_{XO} + n_{OX} + n_{XX}$$
$$= n_0 + 2n_1 + n_2$$

where $n_k$ counts the number of membrane-associated proteins with $k$ binding sites filled. Generally, when a protein has $N$ binding sites, we have

$$[\text{Complex}] = \sum_{k=0}^{N} \binom{N}{k} n_k$$

We write down the detailed balance equation for the protein association/dissociation reactions:

$$n_d k_{mb} = n_0 k_{mu}$$
$$\frac{n_d}{n_0} = \frac{k_{mu}}{k_{mb}} = K_{d,mem}$$

where we have defined the microscopic intrinsic membrane affinity $K_{d,mem}$. The detailed balance equation for the lipid binding/unbinding reactions are:

$$n_k n_{lipid} n_{lb} = n_{k+1} k_{lu}$$
$$n_{lipid} \frac{n_k}{n_{k+1}} = \frac{k_{lu}}{k_{lb}} = K_{d,lip}$$

for all $k = 0,...,N-1$, where we have defined the microscopic intrinsic lipid affinity $K_{d,lip}$. $K_{d,lip}$ varies depending on the protein and the type of lipid (PI(3,4)P$_2$ or PI(4,5)P$_2$). Combining all previous equations, we end up with the expression

$$\sum_{k=0}^{N} \binom{N}{k} \left(\frac{n_{lipid}}{K_{d,lip}}\right)^k = \frac{K_{d,mem}\, n_{lipid}}{K_D\, V_{cell}} \quad (1)$$

This permits the following approach to parametrize the binding model from experimental measurements of $K_D$:

1. Define a simulation cell height $h$ to be used in the simulations, arriving at a cell volume $V_{cell}$.
2. Define a microscopic intrinsic protein-membrane affinity by setting $K_{d,mem}$.
3. Calculate $K_{d,lip}$ by solving equation (1) using the experimental concentration of $n$ lipid and the measured value of $K_D$.

For proteins with one binding site (CALM, Others), equation (1) has an explicit solution:

$$K_{d,lip}^{(1)} = \frac{n_{lipid}}{\frac{n_{lipid}\, K_{d,mem}}{K_{D,Calm}\, V_{cell}} - 1}$$

For proteins with two equal binding sites (SNX9, FCHo 2), Equation (1) has the

following solution:

$$K_{d,lip}^{(2)} = \frac{n_{lipid}}{\sqrt{\frac{n_{lipid}\ K_{d,mem}}{K_D\ V_{cell}}} - 1} \quad (2)$$

For AP-2, the situation is more complex. Experimentally, there is evidence that AP-2 has four binding sites, but these are not equally strong but rather there are two conformations[64] associated with lipid-binding affinities $K_{d,lip}^{+}$ and $K_{d,lip}^{-}$, respectively. Working out the lipid configuration for this case eventually yields the equation:

$$1 + 2\frac{n_{lipid}}{K_{d,lip}^{+}} + 2\frac{n_{lipid}}{K_{d,lip}^{-}} + 2\left(\frac{n_{lipid}}{K_{d,lip}^{+}}\right)^2 + 2\left(\frac{n_{lipid}}{K_{d,lip}^{-}}\right)^2$$
$$+ 8\frac{n_{lipid}^2}{K_{d,lip}^{+}K_{d,lip}^{-}} + 10\frac{n_{lipid}^3}{\left(K_{d,lip}^{+}\right)^2 K_{d,lip}^{-}} + 10\frac{n_{lipid}^3}{K_{d,lip}^{+}\left(K_{d,lip}^{-}\right)^2} \quad (3)$$
$$+ 20\frac{n_{lipid}^4}{\left(K_{d,lip}^{+}\right)^2 \left(K_{d,lip}^{-}\right)^2} = \frac{K_{d,mem}\ n_{lipid}}{K_D\ V_{cell}}.$$

To solve this formula for $K_{d,lip}^{+}$ and $K_{d,lip}^{-}$, we have to use additional information, compared to the equal binding site model in equation (1), since we have now one additional unknown that renders our formula underdetermined without additional experimental input. Fortunately, in the case of AP-2, such additional input is available: The macroscopic dissociation constant $K_D$ for AP-2 is known for the case where all four binding sites are operational[64]. We will refer here to that value by $K_D^{++--}$. In addition, the dissociation constant $K_D$ for a mutant where the two weak binding sites have been mutationally inactivated is known[63]. Using this $K_D$ in the two-binding site model equation (2), we obtain the dissociation constant per strong binding site, $K_{d,lip}^{+} = K_{d,lip}^{(2)}$. Secondly, we can use $K_{d,lip}^{+}$ together with the full $K_D$ of AP-2 where all four binding sites are active in equation (3) and solve for the sole unknown variable $K_{d,lip}^{-}$.

Protein-membrane association parameters were computed using the model described above with published liposome-binding experiments. In these experiments, molecules in the solution bind to a membrane and eventually bind different kinds of lipids. For all proteins but CALM, we assumed the macroscopic $K_D$ of 7.6 μM of AP-2 for binding to PI(4,5)P$_2$ (ref. 64). For CALM, we used 5.8 μM (ref. 65). In our binding model, we assumed CALM and Others to have one binding site, FCHo2 and SNX9 to have two equal binding sites, and AP-2 to have two strong and two weak binding sites[64]. The on-rate of lipid binding was set to 10,000 s$^{-1}$. The cytosolic protein concentrations were set such that the protein numbers found under the clathrin coat in the steady state before the PI(3,4)P$_2$ increase matched the copy number of endocytic proteins found in isolated endocytic CCVs, as described in a recent quantitative proteomic study[42]. Several other parameters (for example, absolute values of rates) are free parameters that do not affect the steady states. The numerical values of the binding model parameters used are given in Supplementary Table 1.

**Simulation setup.** The model is parametrized to have a steady state with absolute protein copy numbers determined from proteomic studies and uses published lipid-binding affinities of the major PI-binding endocytic proteins (Supplementary Table 1). Simulations use discrete time steps of 10 μs and are first run for 20 s before production of PI(3,4)P$_2$ in order to establish a steady state, and then for another 40 s during PI(3,4)P$_2$ production, in order to investigate SNX9 recruitment.

The simulation starts with a concentration of 13,750 free PI(4,5)P$_2$ molecules per μm$^2$, uniformly distributed over the simulation area (about three lipid molecules per simulation cell). This number derives from the measured total concentration of PI(4,5)P$_2$, which has been estimated at 30,000 total PI(4,5)P$_2$ molecules per μm$^2$ (equalling a concentration of 30 μM if distributed freely in the cytosol), of which between 1/2 and 2/3 are bound to abundant PI(4,5)P$_2$-binding proteins such as MARCKs, Ras and others[44,66,67]. This free PI(4,5)P$_2$ concentration is in the upper range of reported values, as other studies have reported even lower values[68]. In the reference parameter set, the PI(4,5)P$_2$ concentration remains constant over the simulation since PI(4)P 5-kinases are absent from CCPs[45,69] (control simulations model PI(4,5)P$_2$ depletion). The time-independent PI(4,5)P$_2$ concentration is a conservative assumption as several PI(4,5)P$_2$-specific phosphatases have been shown to be present in endocytic CCPs[28,70], potentially resulting in progressive PI(4,5)P$_2$ hydrolysis. The scheme is different for PI(3,4)P$_2$: the simulation runs for 20 s with zero PI(3,4)P$_2$ to establish a steady state. Then, PI(3,4)P$_2$ is produced by 33 PI3KC2α at the coat with the experimentally determined turnover number of purified recombinant PI3KC2α (Fig. 1f; $k_{cat} = 4$ s$^{-1}$; $V_{max} = 1,803.6$ nmol min$^{-1}$ mg$^{-1}$) and lifetime[29] at CCPs. PI(3,4)P$_2$ lipids are only produced at clathrin terminal domains, because PI3KC2α binds clathrin at this site and this binding is required for its membrane recruitment and activity[42,46,58,64,65]. To analyse whether the results of the simulation are robust with respect to different parameter changes, the model was validated for a total of 162 different parameter sets. The results, proving the robustness of the model, are reported in Supplementary Table 2 and Fig. 2a,c–f.

**iPRD model.** While the reaction-diffusion model described above is well suited to calculate time-dependent changes of the protein populations for the long timescale of 60 s, it does not resolve the fine details of molecular crowding sufficiently well to predict spatial distributions reliably. Thus, a complementary iPRD model was simulated with the ReaDDy software[51,53]. An unconstrained 'dome model' and a 'constrained model' are set up with 180 Clathrin molecules/540 N-terminal domains as shown in Fig. 4 (see also Supplementary Fig. 2). The clathrin coat has 150 nm diameter. Based on CCP endocytic protein counts[42], we randomly allocated 180 AP-2, 180 CALM, 49 ARH, 38 NECAP1, 33 PI3KC2a, 32 Hip1, 32 HRB, 32 Hip1R, 22 SNX9, 27 Dab2, 27 Necap2, 10 FCHo2 dimers and 20 Epsin molecules to clathrin binding sites at the CCP before the PI(3,4)P$_2$ increase. The number of SNX9 molecules was chosen rather low within the range of sampled parameter settings (compare Supplementary Tables 1 and 2) as the crowding-induced localization of SNX9 to the CCP neck becomes more expressed with increasing protein numbers, that is, a conservative number was chosen. The remaining endocytic proteins quantified in the proteomic study[42] are known to interact with both clathrin and AP-2 (ref. 3), and are thus added to random positions around the already placed AP-2 proteins. Volume exclusion is modelled by representing every protein with a sphere with a radius corresponding to its crystallographic structure or derived from its molecular mass (Supplementary Table 3). Particle overlaps were removed by performing an energy minimization. Informed by the master-equation simulation results (Fig. 2), we inserted 22 SNX9 molecules under the coat and then placed 20 newly PI(3,4)P2 recruited SNX9 molecules into this model by attempting to place them at surface positions generated uniformly at random. A placement attempt was accepted if it did not overlap with existing endocytic proteins. The mean SNX9 placement probabilities ± s.d. using six independently generated CCP protein configurations are depicted in Fig. 4d,i. A single representative realization of placing 20 SNX9 molecules into a given topology is depicted in Fig. 4e. The local mobility of newly associated SNX9 molecules was probed in short Brownian dynamics simulations[71] of 100 ms with the simulation software ReaDDy[51] using the GPU-based version ReaDDy-MM[53]. The diffusional motion of SNX9 molecules was modelled with Browian dynamics, integrated with an explicit Euler scheme. The SNX9 diffusion constant was taken to be 1.5 μm$^2$ s$^{-1}$. Particle repulsion was modelled via repulsive harmonic soft-core potentials between particles. In simulations shown in Fig. 4f,g, an additional potential was introduced to model SNX9–SNX9 interactions. SNX9 molecules are interacting head to head, such that they tend to form horizontally aligned oligomers on the CCP neck, due to the preferred curvature of the BAR domain[72]. The following potential has been derived to capture this behaviour:

$$U(\Delta p) = \begin{cases} -dw(\Delta p) + \frac{1}{2}k(r - r0)^2, & r < r0 \\ \left(-d + \frac{2d(r - r0)^2}{l^2}\right)w(\Delta p), & r \geq r0\ and\ r < \frac{l}{2} + r0 \\ -\frac{2d(r - r0 - l)^2}{l^2}w(\Delta p), & r \geq \frac{l}{2} + r0 \\ 0, & r > r0 + l \end{cases},$$

with SNX9–SNX9 collision distance $r0$, the vector between two particle positions $\Delta p = p_1 - p_0$, the distance between the particles $r = |\Delta p|$, the SNX9–SNX9 repulsion force constant $k$, the potential influence length $l$, the potential depth $d$ and the weighting in $z$ direction

$$w(\Delta p) = \cos\left(\frac{\pi \Delta p_z}{2r}\right)$$

The potential is plotted for parameters $r0 = 6.6$nm, $l = 3$ nm and $k = 10$ k$_B$T and $d = 10$ k$_B$T in Supplementary Fig. 2a. Simulations with different settings of $d$ are shown in Supplementary Fig. 2c. We find that the conservative parametrization of $d = 10$ k$_B$T, corresponding to the formation of two salt bridges[54], are sufficient to drive formation of linear SNX9 assemblies that become rings at narrow necks, and use this setting in all subsequent simulations.

**Antibodies.** See Supplementary Table 4.

**siRNAs.** All siRNAs used in this study were 21-mers or 23-mers including 3′-dTdT overhangs and were custom synthesized by MWG. SiRNA were against dynamin 2 (5′-gca actgaccaaccacatc-3′), PI3K C2α (5′-gcacaaacccaggctattt-3′), SNX18 (5′-caccgacgagaaagcctggaa-3′), clathrin heavy chain (5′-atccaattcgaa-gaccaattt-3′). The siRNA against SNX9 (5′-ggacagaacgggccttgaa-3′) targets the 3′-UTR of the SNX9 mRNA. The scrambled control siRNA used in all experiments corresponds to the scrambled 2 adaptin sequence (5′-gtaactgtcggctcgtggt-3′).

**Plasmids.** The plasmid coding for rat dynamin 2 K44A inserted into peGFP-N1 (pDnm2-K44A-eGFP) was obtained from Addgene (22301) and pcDNA4-V5/His-rat Dynamin2 K44A was a kind gift of Kohji Takei[73]. siRNA resistant pDnm2-GFP was previously described[74]. pEGFP–PI(3)KC2α encoding human full-length PI(3)KC2α and the siRNA resistant plasmid were described previously[29]. eGFP-CLC was inserted into pcDNA3 (gift from M. Krauss). pEGFP-SNX9 encoding human SNX9 and the mutants RYK and K267N K327N were described previously[29]. pEGFP-SNX9 dSH3 lacking amino acids 1–67 was constructed using PCR cloning. Mutations in pEGFP-SNX9 that are F208A/F211A and K522A/K528A (ref. 62) were introduced using phusion mutagenesis (Thermo).

piRFP-SNX9 constructs and mutants were created by transferring SNX9 fragments into piRFP using BamH1/Not1. All plasmids were verified by ds-DNA sequencing.

**Kinetic parameters of PI3KC2α.** Recombinant ΔN PI3KC2α was purified to near homogeneity from baculovirus-infected Sf21 cells. ADP-ATP conversion curves were measured in 384-well format using the ADP-Glo kinase assay (Promega) in the presence of a total concentration of nucleotides of $100\,\mu M$. PI 3-kinase activity was measured by the ADP-Glo assay (Promega) in kinase buffer (5 mM HEPES, pH 7.2, 25 mM KCl, 2.5 mM MgOAc, 150 mM KGlu, $10\,\mu M\ CaCl_2$, 0.2% (w/v) CHAPS) using $400\,\mu M\ PI(4)P$ (Avanti) and $100\,\mu M$ ATP as substrates. Kinase reactions performed at 37 °C for 15 min using serial dilution of the enzyme. Background subtracted data (in the absence of enzyme) were analysed by linear regression analysis using the standard ADP/ATP conversion curve.

**Purification of SNX9 PX-BAR.** The cDNA encoding the PX-BAR domain (aa 204–595) of SNX9 was inserted into pGEX4T-1. The protein was expressed in *E. coli* BL21 (DE3) for 20 h with 0.05 mM IPTG induction at 20 °C. The protein was purified using GST-Bind resin (Novagen), followed by on-bead cleavage of the GST tag by thrombin, ion exchange, and Superdex S200 16/30 gel filtration chromatography (GE Healthcare). The peak SNX9 fractions were pooled and concentrated to $5\,mg\,ml^{-1}$. The protein purity was verified with SDS-polyacrylamide gel electrophoresis and Coomassie blue staining.

**Biosensor experiments.** Liposomes were prepared using the following method[75]: for basic liposomes 80% PC, 20% PE (w/w), lipids (stored in chloroform/methanol, nitrogen atmosphere) were mixed and dried under SpeedVac without heating, then rehydrated with $150\,\mu l$ of 0.3 M sucrose for 1 h. The volume was subsequently adjusted to 1 ml using H2O and the liposomes sedimented by centrifugation at 20,000 r.p.m. for 1 h at 4 °C and resuspended in $500\,\mu l$ 20 mM HEPES, (pH 7.4). Twenty-three times passage through a 100 nm membrane yielded unilamellar liposomes. For $PIP_2$ containing liposomes, 2% of PC was replaced with $PI(3,4)P_2$ or $PI(4,5)P_2$. The L1 chip (Biacore T200, GE Healthcare) was used to immobilize liposomes. The chip surface was primed with running buffer (20 mM HEPES, pH 7.5, 200 mM NaCl and 5 mM DTT), followed by three injections of 20 mM CHAPS for 30 s at a flow rate of $10\,\mu l\ min^{-1}$, and liposome immobilization (1:5 to 1:10 dilution in running buffer). Non-bound liposomes were removed with two injections of 50 mM NaOH for 10 s at a flow rate $30\,\mu l\ min^{-1}$ to reach a final level of 6,000–10,000 RU. Purified SNX9 (PX-BAR) at concentrations ranging from $8\,\mu M$ to 31 nM was used as an analyte at a flow rate $30\,\mu l\ min^{-1}$ with 360 s injection and 360 s dissociation. The immobilized liposome surfaces were regenerated with 50 mM NaOH injection twice for 20 s at a flow rate $30\,\mu l\ min^{-1}$. Data were analysed using the manufacturer supplied software.

**Cell lines.** Cos7 and HeLa cells were from ATCC and not used beyond passage 30 from original. Stable EGFP–clathrin expressing Cos7 were previously described[76] and maintained in presence of $4\,\mu g\,ml^{-1}$ Puromycin. Genome-edited SK-MEL-2 DNM2$^{en-1}$-eGFP/CLTA$^{en-1}$-RFP, DNM2$^{en-all}$-eGFP and CLTA$^{en-all}$-RFP genome-edited cells[77] were cultured in DMEM/F-12 and 10% FBS (Hyclone). All cell lines were routinely tested for mycoplasm contaminations on a monthly basis.

**siRNA and plasmid transfections.** HeLa and Cos7 cells were transfected with siRNA using Oligofectamin (Life Technologies) according to the manufacturer's protocol. SK-MEL-2 cells were transfected with siRNA using siRNA Max (Life Technologies) according to the manufacturer's protocol. To achieve optimal knockdown efficiency, two rounds of silencing were performed on day 1 and day 2. For transient overexpression of proteins in knockdown cells, plasmids were transfected together with the second round of siRNA using Lipofectamin2000 (Life Technologies) according to the manufacturer's protocol. Cells were analysed 72 h after the first siRNA transfection. For transient overexpression of proteins in untreated cells, plasmids were transfected 24 h prior to analysis using Lipofectamin2000 (Life Technologies).

**Immunocytochemistry.** Cultured cells seeded on Matrigel-coated coverslips were fixed for 10 min with 4 % PFA, washed two times with PBS, permeabilized in blocking solution (30% goat serum, 20 mM $Na_2HPO_4$ pH 7.4, 0.3 % Triton X-100, 100 mM sodium chloride) for 30 min and incubated with primary antibodies diluted in blocking solution for 1 h. After three washes with PBS, secondary antibodies diluted in blocking solution were incubated for 1 h. Protein stainings were routinely analysed and quantified using a laser scanning confocal microscope (Zeiss 710) or an epifluorescence microscope (see below). For quantification of SNX9-GFP localization in HeLa cells, cells with either diffuse or clustered localization were counted.

**Sample preparation for dSTORM imaging.** For super-resolution imaging, cells were grown on acid-cleaned coverslips ($170\pm5\,\mu m$, Carl Roth GmbH). Cytoplasmic proteins from cells expressing Dnm2-K44A-GFP or eGFP-tagged CLC, PI3KC2α or SNX9 were extracted in preextraction buffer (25 mM Hepes,

25 mM KCl, 2.5 mM $MgCl_2$, 0.05% (w/v) saponin, pH 7.4) for 1 min at 37 °C before fixation in 4% PFA. Alternatively, cells expressing Dnm2-K44A-GFP were fixed in methanol at $-20$ °C for 5 min. After fixation with 3% PFA for 15 min at room temperature, cells were washed three times with PBS at room temperature. Immunocytochemistry was done as described above except the secondary antibodies conjugated to CF647 or CF680 (Biotium) were used at $10\,\mu g\,ml^{-1}$. Fluorescent beads (100 nm Tetraspek or 40 nm dark Red, Life Technologies) were diluted in Poly-L-lysine and bound to the coverslips during 5 min at room temperature. Samples were washed three times in PBS and mounted in imaging buffer (50 mM Tris/HCl, 10 mM NaCl, pH 8) with 10 mM β-mercaptoethylamine (MEA, Sigma 30070) and an oxygen scavenging system consisting of $0.5\,mg\,ml^{-1}$ glucose oxidase (Sigma G2133), $40\,\mu g\,ml^{-1}$ catalase (Sigma C100) and 10% (w/v) glucose. Stained samples were mounted on slides providing a $100\,\mu l$ spherical void (Carl Roth, H884.1) with imaging buffers, sealed with picodent twinsil and imaged within 8 h.

**dSTORM imaging and analysis.** We used the μManager[78] controlled, custom-built SD-dSTORM system[79] based on a custom laser combiner and TIRF illuminator system, a Nikon Eclipse Ti research microscope, an emission splitter (OptoSplit II, Cairn Optics) and an EMCCD (DU-897E, 512 × 512, Andor Instruments). For activation and excitation, we used lasers lines at 405 nm (50 mW, Qioptiq) and 643 nm (150 mW, Toptica), respectively. Sample illumination occurred in wide-field mode with excitation intensities $0.36\,kW\,cm^{-2}$ at 643 nm. The laser beams were reflected off a quadband dichroic (Di01-R405/488/561/635). We collected the emission of dyes excited with 643 nm using a longpass filter (BLP01-635R). All filters were from AHF Analysetechnik. The combination of a × 100 1.49 numerical aperture objective and a × 1.5 optovar (Nikon) and the EMCCD with a pixel size of $16 \times 16\,\mu m^2$ resulted in pixel sizes of $105 \times 105\,nm^2$ in the image. Initially, the fluorescent dyes were switched off until single-molecule blinking was detected. Image acquisition was performed at a frame rate of 33 Hz, typically for 20,000 frames. Lower labelling density was compensated by reactivation of CF647 with low intensities of ultraviolet light (405 nm, $0.05\,W\,cm^{-2}$). Dual colour spectral demixing (SD) dSTORM imaging was performed using CF647 and CF680. A custom software[80] was used to identify corresponding localization pairs in the drift-corrected output of rapidSTORM file. A localization pair was identified in the short- and long-wavelength side of the emission splitter, using the emission splitter's geometrical offset. A pair is counted if it is inside a 158 nm search radius in x/y. For 3D imaging, an astigmatic lens ($f = 1\,m$, Thorlabs LJ1516RM-A) was inserted into the short channel (Em <700 nm) that was used for image reconstruction. Single molecules were localized with the open source software rapidSTORM 3.2 (ref. 81). RapidSTORM performs a Gaussian fit (Levenberg–Marquardt parameter estimation) to each single-molecule signal using a fixed PSF-FWHM (point spread function's full width at half maximum) of 300 nm for CF647. The fitting window radius was set to 600 nm and the minimum spot distance to 5 pixels. The minimum threshold of signal intensity (that is, the total intensity under the 2D Gaussian fit) was set to 1000 ADC (analogue/digital counts). The 'two-kernel improvement' was used to prevent the detection of multiple single-molecule localizations appearing in the same frame within a radius of 600 nm. The rapidSTORM output is a list of x,y,z coordinates with subpixel localization accuracy, the total intensities (integrated Gaussian fit) and frame number of each detected single molecule. Drift correction was performed by a custom-written program based on Python 2.7 that uses fluorescent beads as fiduciary markers. The accuracy of this correction was calculated from the s.d. $\sigma$ of the distances of the corrected bead localizations as $2.35\sigma$. Typically, we achieved a drift correction accuracy of 4–5 nm. The final dSTORM image of CF647-labelled structures had an image resolution of 20–24 nm and was reconstructed using 10 nm pixel size. The final 3D SD dSTORM image of CF647/CF680-labelled structures had an lateral and axial localization precision of 25 and 60 nm, respectively (similar to ref. 82) and was reconstructed using $10 \times 10 \times 25$ nm voxel size.

**Superimposition and analysis of single-molecule data.** The CCPs for averaging were selected based on the presence of bright signal from Dnm2 K44A-eGFP in the widefield image using a custom ImageJ script that created ROIs with the centre at the position of the mouse click. Particle averaging was performed using a custom program written in Python selected the corresponding localization data from the rapidSTORM list of all localizations. The superimposed localizations from all selections were plotted together using rapidSTORM with a pixel size of 10 nm. Distance measurement were performed on dual colour 3D SD dSTORM images. The x–y–z distance of the median position of the two channels from arbitrary choosen CCP was computed using a custom program written in Python. Phyton scripts for drift correction, grouping, centre-averaging and distance measurements of dSTORM images can be provided upon request.

**Tf uptake and surface labelling.** Cells seeded on Matrigel (BD Biosciences)-coated coverslips were serum-starved for 1 h and used for Tf uptake and subsequent surface labelling. For quantitative Tf uptake, cells were treated with $25\,\mu g\,ml^{-1}$ Tf-Alexa568 (Life Technologies) for 10 min at 37 °C. After being washed twice with ice-cold PBS, cells were acid washed at pH 5.5 (0.1 M sodium acetate, 0.2 M NaCl) for 1 min on ice to remove surface bound Tf, followed by two

times washing with ice-cold PBS. For subsequent TfR surface labelling, cells were further incubated with 25 µg ml$^{-1}$ Tf-Alexa647 (Life Technologies) for 45 min at 4 °C to block endocytosis, washed three times with ice-cold PBS and fixed with 4% PFA for 45 min at room temperature.

Tf uptake and surface labelling were analysed using an Epi-fluorescence microscope (Nikon Eclipse Ti microscope) equipped with optimal filters (eGFP filter set: F36-526; TexasRed filter set: F36-504; Cy5 filter set: F46-009; Dapi filterset: F46-000, AHF Analysentechnik), a ×40 oil-immersion objective (Nikon) a sCMOS camera (Neo, Andor), a 200 W mercury lamp (Lumen 200, Prior) and operated by open-source ImageJ-based micromanager software. Internalized and surface bound Tf was quantified in randomly selected eGFP positive cells and normalized to the cell area using automated software (ImageJ).

**TIRF microscopy.** TIRF microscopy was performed using a Nikon Eclipse Ti microscope, equipped with an incubation chamber (37 °C), a ×60 TIRF objective (Apo TIRF 1.49NA, Nikon), a sCMOS camera (Neo, Andor), a 200 W mercury lamp (Lumen 200, Prior), a four-colour TIRF setup (laser lines: 405, 488, 568, 647 nm) an appropriate dicroic mirror(Di01-R405/488/561/635), filter (FF01-446/ 523/600/677) and operated by open-source ImageJ-based micromanager software. Cos7 eGFP-CLC or SK-MEL-2 cells were seeded on Matrigel-coated coverslips 24 or 4 h prior to imaging, respectively. Imaging was performed at 37 °C in live cell imaging buffer (Hank's balanced salt solution +Ca$^{2+}$ +Mg$^{2+}$, 5% FCS) over 180 s (Cos7) or 300 s (SK-MEL-2) at an imaging rate of 0.5 Hz. For all time-lapse movies, the 488 nm channel was acquired before the 568 and 647 nm channel. Quantitative analysis of Dnm2-GFP levels at CCPs was performed from TIRF movies of SK-MEL-2 Dnm2-GFPen-1/mRFP-CLCen-1. Quantitative analysis of GFP-PI3KC2α, GFP-SNX9 and Dnm2-GFP levels at CCPs was performed from TIRF movies of SK-MEL-2 mRFP-CLCen-all after two rounds of siRNA knockdown and transfection with siRNA resistant plasmids. The maximal fluorescence intensity from arbitrary selected CCP was background corrected by substracting the median background intensity. The corrected fluorescence intensities were compared to known molecule numbers for Dnm2-GFP at CCPs[35] with 26 molecules corresponding to 5,500 intensity units. Clathrin and dynamin 2 dynamics were determined manually from arbitrary selected CCP present in the middle frame of a movie. Average traces for Clathrin, SNX9 and Dnm2 were obtained from manually aligned sequences to the disappearance frame of the dynamin 2 signal. Representative kymographs were chosen over 180 or 300 s along a line of 300 pixels in length.

**Immunoblot analysis.** Knockdown of SNX9 and 18 was routinely verified by western blot analysis. Cells were collected 72 h after first siRNA transfection in lysis buffer (20 mM HEPES, pH 7.4, 100 mM KCl, 2 mM MgCl$_2$, 1 mM PMSF, 0.1% protease inhibitor cocktail, 1% Triton X-100) and incubated on ice for 30 min, followed by centrifugation at 17,000g for 20 min at 4 °C. Protein concentrations were determined via Bradford assay and 10–15 µg protein were loaded onto a 4–15% acrylamide gel (BioRad) for SDS-polyacrylamide gel electrophoresis followed by immunoblotting. Western blot development was done using a LI-COR Odyssey Fc imager. Only experiments with high knockdown efficiency (>85%) were analysed. See Supplementary Fig. 6 for uncropped western blots.

**Electron microscopy.** Scrambled or siSNX9/18 siRNA transfection was performed as described in Cos cells that were grown in 6 cm plastic dishes. Cells were fixed with 2% glutaraldehyde in PBS. After rinsing in fresh PBS, cells were mechanically detached by scratching, pelleted and embedded into gelatin. Following osmification with aqueous 1% osmium tetroxide samples were stained en bloc with 1% aqueous uranyl acetate and embedded in epoxy resin. Sections were viewed with Zeiss 910 transmission electron microscope and micrographs were taken along the cell perimeter at ×20,000. For morphometric analysis images were combined in order to reconstruct the perimeter of a cell and the number of clathrin-coated intermediates in up to 1 µm distance from the plasma membrane was counted. Cell profiles (19–33) were analysed per condition

**Statistical analysis.** For analysis of experiments comprising multiple independent experiments (n), statistically significant estimates for each sample were obtained by choosing an appropriate sample size, correlating to 15–30 images per condition for microscopy-based quantifications. Cells were chosen arbitrarily according to the fluorescent signal in a separate channel, which was not used for quantification. All statistical tests were performed using the two-tailed, paired or unpaired t-test or one-way ANOVA with Dunnett's Multiple Comparison Test.

**Data availability.** All relevant data is available from the authors. The iPRD simulation code ReaDDy is available via BSD clause 2 licence, either via http://readdy-project.org (download and tutorials) or https://github.com/readdy (source code). The RDME simulation code is available on request from the authors.

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

## Acknowledgements

We thank Dr David Drubin (UC Berkeley, CA, USA) for the kind gift of genome edited SK-MEL-2 cells, Dr Sven Carlsson (University of Umea, Sweden) for anti-SNX9 antibodies, Dr Oliver Daumke for critical reading and comments on the manuscript and Martina Ringling, Maria Mühlbauer and Silke Zillmann for technical assistance. The study was supported by grants from the German Research Foundation to F.N. (SFB958/A04 and SFB1114/C03), J.Schm. (SFB958/Z02), and V.H. (SFB958/A07) and European Commission to F.N. (ERC StG 'pcCell') and J.Schö. (Marie Skłodowska-Curie IOF). Additional data are available in the Supplementary Materials.

## Author contributions

M.L., Y.P., W.-T.L. and J.Schm. performed experiments; J.Schö. and A.U. performed simulations; J.Schö., G.L. and A.U. developed code; J. Schö. and F.N. developed theory; J.Schö., J.Schm., M.L., V.H. and F.N. designed research; J.Schm., V.H. and F.N. wrote the manuscript.

**Additional information**

**Competing interests:** The authors declare no competing financial interests.

