## [Peer Review File · Nature Communications]

Reviewers' Comments:

Reviewer #1 (Remarks to the Author)

The molecular mechanisms underlying membrane constriction and fission during endocytosis are incompletely understood. In the present study, the authors combined simulations with biochemical and cell biological experiments to unravel the mechanism by which PX-BAR domain protein SNX9 is recruited to the endocytic vesicle neck and how its function is linked to dynamin-mediated fission.

Majority of the experiments presented are of good technical quality and the study provides interesting new information about interplay between local PI(3,4)P₂ synthesis, SNX9, and dynamin. However, there are several points that should be addressed to further strengthen the manuscript.

Major points:

1. I have some concerns about the parameters that were chosen for the simulations. First, why only a subset of phosphoinositide-binding endocytic proteins was included, and for example amphiphysin (or other important endocytic PIP₂-binding N-BAR domain proteins) was not included in simulations? Second, I do not understand how it is possible to determine the exact K_d-values for membrane interactions of N-BAR and F-BAR domain proteins (see Fig. 1c, supplementary Fig. S1 and supplementary Table SI), without concrete information about the stoichiometry of binding. BAR/F-BAR domains bind phosphoinositides through large positively charged surfaces, induce phosphoinositide clustering, and thus most likely interact simultaneously with several phosphoinositide molecules (i.e. each BAR/F-BAR dimer is likely to interact with more than 2 phosphoinositide molecules). Therefore, the issue of binding stoichiometry should be at minimum discussed in the context of simulations presented in the manuscript.
2. Based on the data presented in Figs. 6 and 7, SNX9 (or SNX18) is not essential for constriction of CCPs. Do SNX9 and other BAR domain proteins (e.g. amphiphysin) have overlapping roles in CCP constriction and dynamin recruitment?
3. Does SNX9 limit the lateral diffusion of PI(3,4)P₂ as shown recently for yeast endocytic BAR domain proteins (Zhao et al., Cell Rep., 2013)? This parameter would be important to incorporate to the model/simulations or at least discuss as a possible limitation of the simulations.
4. The authors should provide a more balanced view on the topic by also discussing mechanisms by which various BAR superfamily proteins contribute to endocytosis in yeast (e.g. Kishimoto et al., PNAS, 2011), and the links between SNX9 and actin dynamics (Shin et al., JCS, 2008; Gallop et al., PNAS, 2013).
5. In the 'Introduction' (page 3), the authors state that actin functions as a 'stabilizing protein' in endocytosis. This is misleading, because several studies on yeast and animal cells have demonstrated that coordinated actin polymerization provides force for generating endocytic invaginations (e.g. Kaksonen et al., Cell, 2003; Messa et al., eLife, 2014). This should be clarified in the manuscript.
6. Based on Fig. 3, wild-type SNX9 does not form clusters in control cells (but forms bright clusters in dynamin 2 knockdown cells). This result should be explained/clarified in the manuscript text, because in its current form it seems confusing and controversial e.g. with the data presented in Fig. 4.

Minor points:

1. The panels in Fig. 1 do not match with the corresponding figure legend.
2. Scale bars should be included in Fig. 4.

Reviewer #2 (Remarks to the Author)

The article by Schöneberg et al represents an interesting attempt to recover, computationally, and by super-resolution imaging the sequence of the protein organization events leading to formation of endocytic neck. According to the major idea of the model, the background for the process includes enzymatic conversion of [PI(4,5)P₂] into [PI(3,4)P₂] underneath the clathrin coat, [PI(3,4)P₂] lateral diffusion into the membrane region surrounding of the coat, and recruitment of BAR domain protein SNX9. The model predicts a localized SNX9 binding along the rim of the coat. The model suggests that formation of a ring-like region enriched in SNX9 around the coat generates additional bending of the endocytic neck and assists dynamin in driving the neck fission.

The study is very systematic, thorough and represents a great example of a concerned computational-experimental study, where experimental knowledge determines the values of parameters of the theoretical model, while the predictions following from the model solution are checked by specifically designed experiments.

I think the article deserves publication in Nature Communications after some revision of the presentation, which, at this stage, seems a little confusing.

Specifically:

1. It will help if the authors describe in simple terms and straightforwardly the scenario predicted computationally and explain the physics behind it. Such description may already exist in the paper, but it is spread over the text and is, therefore, difficult to summarize. Specific questions would be:

(i) why does the model predict concentration of SNX9 within a limited region near the coat edge rather than SNX9 spreading all over the membrane outside the coat? If this is due to SNX9 oligomerization, which builds around the coat a fence impermeable for diffusion of newly synthesized SNX9 molecules, what happens with the latter? How is the steady state reached in this case?

(ii) Is the time sequence of protein recruitment – SNX9 before dynamin – the result of the model or an assumption?

2. I got an impression, that the computations were performed for a flat rather than curved membrane. If this is the case, it should be discussed how the shape of a real membrane neck can affect diffusion of the membrane-bound proteins and whether this may influence the model results.

3. The suggestion of the model, that SNX9 concentration can shape the membrane neck to the extent implied by the experimental results should be supported, quantitatively, by structural features of this specific protein. Indeed, the curvatures produced and sensed by different BAR domains vary in a broad range from about 1/10nm for endophilin and amphiphysin, through very low values for F-BARs, to negative values for I-BARs. Hence, to justify the model a number is needed for SNX9.

4. Since the authors conclude that SNX9 promotes the fission process, some thoughts are needed on a specific mechanism of this effect. If no reasonable idea could be suggested at this stage, I suggest to downplay in the paper the issue of membrane fission.

Reviewer #3 (Remarks to the Author)

The manuscript by Schöneberg et al. is interested in the specific recruitment of Snx9 to the neck of endocytic pits by the specific production of PI(3,4)P2 by the enzyme PI3KC2. It shows that, even if the PX domain of Snx9 has no preference for one of its 2 possible ligands PI(3,4)P2 and PI(4,5)P2, it is specifically recruited to the pit by PI(3,4)P2 because of competition of other CCPs players also binding PI(4,5)P2. The manuscript is a nice combination of in silico simulations, biochemical measurements to extract relevant parameters, and cell biology experiment to qualitatively validate the model.

However, there are three points that I think should be experimentally or computationally addressed to allow for publication, as they come in conflict with the authors conclusions.

1-Diffusion of PI(3,4)P2, enzymatic activity of PI3KC2 and time recruitment of Snx9

From their simulation, the authors show that the recruitment of Snx9 is perfectly coordinated with the increase of PI(3,4)P2 in the membrane. However in cells, as shown by the authors in their previous paper (Posor et al.), PI3KC2 comes much before Snx9, suggesting that the Snx9 comes long after the start of PI(3,4)P2 production. This discrepancy has to be solved, or explained to support the conclusion of the authors that Snx9 is recruited to the pit by PI(3,4)P2.

In my opinion, it could come from a diffusion problem. When produced by PI3KC2, the PI(3,4)P2 would diffuse very fast out of the pit, not increasing locally its concentration. In this case, it would only be when PI3KC2 activity at the pit reaches a certain threshold, enough to overcome the lipid diffusion, that the local concentration of PI(3,4)P2 would increase, recruiting Snx9. But what is the enzymatic activity of PI3KC2 at the pit?

The authors state that from their enzymatic activity measurement, and the average number of PI3KC2 subunits present in the coat, they can compute the PI(3,4)P2 production. But they don't show any number nor detailed explanation. Going back to Borner et al. JCB 2012, I found that on average, there are 10 copies of PI3KC2 per CCV. Using the enzymatic activity measured by the authors (4 per s), this leads to the production of about 40 PI(3,4)P2 per second. How fast do these 40 lipids diffuses?

The authors simulate 0.12 μm^2 of membrane, which would take about 0.04 s for each lipid to move out this area, since they diffuse (using the value given by the authors) at 3 $\mu\text{m}^2/\text{s}$. That means that on average, there is $40 \times 0.04 = 1.6$ PI(3,4)P2 present in the entire membrane area simulated by the authors with the average enzymatic activity of PI3KC2. Thus diffusion of lipids is clearly limiting, and I request the authors to perform the following simulations/experiments/explanations to address this point.

First, it is not clearly indicated in the paper (I could not find a clear indication), but I guess the authors performed the simulation with a closed area of 0.12 μm^2 , meaning that each molecule added to this area would diffuse fast, but stay in the boundaries of the area. If this is the case, I would suggest that simulations should be run with a change of boundary conditions, in which PIP lipids (only), when they pass the boundary of the simulated domain, are lost, to mimic a case in which a CCP is connected to an infinite membrane, a case more physiologically relevant. This will allow to find the sufficient PI(3,4)P2 production rate to overcome diffusion.

Second, the calculation I did above is pretty gross. In my opinion, as the authors have shown in their previous paper that PI3KC2 accumulate with time in the CCP, I could envision a mechanism in which, at a given time during the accumulation of PI3KC2 at the pit, the production rate of PI(3,4)P2 overcomes the diffusion and starts accumulating PI(3,4)P2 at the pits to recruit Snx9. In this case, it would be expected the recruitment of Snx9 starts way after the accumulation of PI3KC2, as seen in cells. I would thus request the authors to perform simulations with the same PIP losing condition as above, but with the accumulation of PI3KC2 and/or steady increase of

PIP₃,4P₂ production. Overall, I think it would be essential to model the PI3KC2 activity and diffusion of PIPs to solve the discrepancy.

In cells, it would be interesting to calibrate the microscope to estimate the number of PI3KC2 copies per pit with time. This would allow for comparing with the simulations proposed above, and show that the overall production rate of PI₃,4P₂ is sufficient for overcoming the lipid diffusion.

2-the 100 protein threshold.

I have a problem with the "constriction threshold" that the authors have chosen. This is essential because if the value of the threshold (100 copies) is different, all the conclusions of the authors are likely to be wrong. For example, a threshold of 50 copies would completely change the conclusions of figure 2. The rationale for choosing 100 as the constriction threshold seems fairly arbitrary to me. Based on structural arguments, the authors say it is the minimal number of copies to form a full ring around U-shaped pits. However, constriction and membrane deformation (see also point 3) by BAR proteins is a continuous process, and depends on the membrane-bound densities, not on a specific number (see Sorre et al. PNAS 2012, and Simunovic et al. PNAS 2016). Thus constriction will be happening as soon as a single copy will be present. The curvature generated by 80 copies is likely not to be so much different from the curvature generated by 200 copies, which are presented as non-constricted (early stage below the threshold) and constricted state of the neck (late stage above the threshold). Thus, the conclusions of the authors that the accumulation of PI(3,4)P₂ drives a further recruitment of Snx9 above a threshold allowing for constriction seems strongly dependent on the threshold value, is, in my opinion, conceptually wrong. I would instead propose that there is a continuous accumulation of Snx9 at the pit, which may increase curvature, in conjunction with other BAR proteins (see below).

In order to define more rigorously the threshold value, I request the authors to quantify the number of Snx9 copies at the pit by calibrating their microscope. In this case, the threshold value could be set as 50% of the maximal number of copies at the pit. It would be less arbitrary.

Finally, the simulations seem to indicate that the presence of PI(4,5)P₂ is sufficient to drive Snx9 binding in absence of PI(3,4)P₂ (and in presence of competing PI(4,5)P₂ binding partners). I would thus have expected Snx9 to be at least a little bit present from the very start of the CCP formation. However, in cells, Snx9 seems to be present only at the end of the CCP formation, whereas all other partners (PI(4,5)P₂ competing components of the clathrin coat) are present from the beginning. It seems inconsistent to me, and I thus wonder how much the simulations reflect the in vivo situation.

3-constriction mechanism, role of other BAR proteins.

In all simulations, membrane curvature is not coupled to binding (Fig2 is made with flat membranes, and other figures do not allow for curvature change as a function of protein density). For BAR proteins, it is very well described that binding to the membrane is coupled to curvature. This effect is probably as important as the lipid binding effect, and should be taken into account. For example, the fact that Snx9 does not bind to the pit in early stage may not be due to the absence of PI(3,4)P₂, which is unlikely since PI3KC2 is present, but because the pit is not curved enough, not allowing for Snx9 BAR binding. Also, the ring organization discussed in fig4 and supp. figS2 strongly depend on the size of the neck, and thus on this coupling mechanism between binding and curvature. I understand it is not easy to simulate this coupling and that the algorithm written by the authors are not meant to do this, but I think it would be a plus if the authors could at least give us an idea of how the conclusions of figure 4 change with gradual increase of the neck size (only 2 extreme sizes have been tested).

Also, the structure of the BAR domain of Snx9 is very similar to the one of Endophilin and Amphiphysin, which appear at the pit at the same time than Snx9, but however do not bind PI₃,4P₂ specifically. I would have expected that if Snx9 is not present (siRNA), Endo and Amphi would totally compensate for the loss of Snx9. The authors should check the localization of Endo and Amphi in Snx9 siRNA cells, as well as in PI3KC2 siRNA cells. The accumulation of U-shaped

CCPs in those cells suggest that both Endo and Amphi are not recruited, which make it difficult to believe that the non-constricted neck phenotype is solely due to the absence of Snx9...

Reviewer #1:

The molecular mechanisms underlying membrane constriction and fission during endocytosis are incompletely understood. In the present study, the authors combined simulations with biochemical and cell biological experiments to unravel the mechanism by which PX-BAR domain protein SNX9 is recruited to the endocytic vesicle neck and how its function is linked to dynamin-mediated fission.

Majority of the experiments presented are of good technical quality and the study provides interesting new information about interplay between local PI(3,4)P2 synthesis, SNX9, and dynamin. However, there are several points that should be addressed to further strengthen the manuscript.

Response: We thank the referee for these very positive remarks and for highlighting the importance and quality of our work.

1. I have some concerns about the parameters that were chosen for the simulations. First, why only a subset of phosphoinositide-binding endocytic proteins was included, and for example amphiphysin (or other important endocytic PIP2-binding N-BAR domain proteins) was not included in simulations?

Response: We thank the referee for raising this point. In the original manuscript we had focused on the major endocytic proteins by copy number as determined experimentally by Borner et al (2012). However, we agree with the referee that other endocytic PI-binding proteins may be relevant for the computational model and have added these (summed and denoted "other") to the new simulations presented in the **new Figures 2, 4, and Suppl. Tables 1, 3.**

Second, I do not understand how it is possible to determine the exact K_d -values for membrane interactions of N-BAR and F-BAR domain proteins (see Fig. 1c, supplementary Fig. S1 and supplementary Table S1), without concrete information about the stoichiometry of binding. BAR/F-BAR domains bind phosphoinositides through large positively charged surfaces, induce phosphoinositide clustering, and thus most likely interact simultaneously with several phosphoinositide molecules (i.e each BAR/F-BAR dimer is likely to interact with more than 2 phosphoinositide molecules). Therefore, the issue of binding stoichiometry should be at minimum discussed in the context of simulations presented in the manuscript.

Response: We again agree with the referee that it is possible that more than 2 lipids may associate with the PX-BAR domain dimer of SNX9 and that this may affect the determined K_D as now explicitly stated in the legend to Fig. 1. Importantly, however, our binding model described in the SI section *Parametrization of the protein-lipid binding in the RDME model* treats both, unspecific membrane binding due to an intrinsic affinity to an oppositely charged membrane

surface and specific binding of PI lipids to the PI-binding site on the PX domain. Parametrization of this model reveals that the unspecific binding is the minor effect. This is in agreement with the experimental observation that membrane recruitment of SNX9 is completely eliminated by mutation of the PI-binding site on the PX domain for which crystallographic data exist. Note that our binding model takes the number of binding sites explicitly into account. For example, SNX9 comes in dimers, and has therefore two binding sites, which use the same PI-binding affinities. The effective (experimentally measured) binding affinity of the dimer arises from the combination of all possible system configurations (unspecific binding, specific binding with one PI, specific binding with two PIs). We have now clarified this point in the description of the recruitment model (first Results section).

2. *Based on the data presented in Figs. 6 and 7, SNX9 (or SNX18) is not essential for constriction of CCPs. Do SNX9 and other BAR domain proteins (e.g. amphiphysin) have overlapping roles in CCP constriction and dynamin recruitment?*

Response: Thank you! We have analyzed the expression levels and localization of amphiphysin 1 and endophilin A2 in control and SNX9/18-depleted cells. In our **new Suppl. Fig. 5** we report that amphiphysin 1 levels at coated pits are increased in SNX9/18-depleted cells, possibly as a result of compensation that, however, is insufficient to fully rescue loss of SNX9/18 as U-shaped clathrin-coated pits accumulate in these cells (please see Figure 6). We do not see any change in the (limited) colocalization of endophilin A2 with clathrin, confirming that its expression or recruitment to endocytic sites is not affected by loss of SNX9/18. Based on these results we conclude that SNX9/18 are indeed required for constriction.

3. *Does SNX9 limit the lateral diffusion of PI(3,4)P₂ as shown recently for yeast endocytic BAR domain proteins (Zhao et al., Cell Rep., 2013)? This parameter would be important to incorporate to the model/simulations or at least discuss as a possible limitation of the simulations.*

Response: We are grateful to the referee for raising this important point. In response to this point and also to a change of simulation boundary conditions suggested by *referee 3*, we have replaced the previous protein recruitment simulations (Fig. 2), which were carried out within a bounded domain by new simulations, in which PI-lipids could escape the simulation domain but were slowed down to account for the restricted mobility of lipids due to the dense protein coat at the CCP.

Using this model together with experimental data (Zhao et al., Cell Rep. 2013) on the restriction of lipid diffusion by endocytic BAR domain proteins (also refereed to by *referee 1*) we now show in the **new Figure 2** and **Suppl. Tables 1 and 2** that clathrin-coated pit enrichment of PI(3,4)P₂ synthesized by PI3KC2a depends on the slowdown of lateral lipid diffusion by endocytic proteins assembled underneath the clathrin coat. Our new simulations confirm the prediction of *referee 3* that the buildup of a local PI(3,4)P₂ gradient at coated pits depends on the ability of the endocytic coat to limit the lateral diffusion of PI lipids by at least two orders of magnitude, consistent with recent experimental data on lipid mobility under protein coats (Zhao et al., Cell Rep. 2013). These results provide strong additional support to our initial findings and considerably extend the computational model.

4. *The authors should provide a more balanced view on the topic by also discussing mechanisms by which various BAR superfamily proteins contribute to endocytosis in yeast (e.g. Kishimoto et al., PNAS, 2011), and the links between SNX9 and actin dynamics (Shin et al., JCS, 2008; Gallop et al., PNAS, 2013).*

Response: We apologize for not quoting these works properly, which has now been corrected in the revised Ms.

5. In the 'Introduction' (page 3), the authors state that actin functions as a 'stabilizing protein' in endocytosis. This is misleading, because several studies on yeast and animal cells have demonstrated that coordinated actin polymerization provides force for generating endocytic invaginations (e.g. Kaksonen et al., Cell, 2003; Messa et al., eLife, 2014). This should be clarified in the manuscript.

Response: We have amended the text accordingly and added the cited works to the reference list.

6. *Based on Fig. 3, wild-type SNX9 does not form clusters in control cells (but forms bright clusters in dynamin 2 knockdown cells). This result should be explained/clarified in the manuscript text, because in its current form it seems confusing and controversial e.g. with the data presented in Fig. 4.*

Response: We have added additional explanations in the text and have quoted the corresponding work (Ferguson et al, Dev Cell 2009. In brief, the apparent discrepancy is explained by the fact that Fig.3 shows regular widefield images of SNX9 and clathrin in fixed cells, while Figure 4 shows dynamic recruitment of clathrin, dynamin 2, and SNX9 to CCPs via TIRF imaging in living cells. As dynamin 2 and SNX9 are recruited in short bursts to late stage CCPs in control cells only some CCPs are in late stages and recruit SNX9. KD of dynamin 2 causes late-stage arrest, resulting in the accumulation of wild-type (see also Ferguson et al, Dev Cell 2009) but not PI(3,4)P₂ binding defective mutant SNX9 as predicted by our model.

Minor points:

1. The panels in Fig. 1 do not match with the corresponding figure legend.

Response: We have corrected the legend to Figure 1 and apologize for the error.

2. Scale bars should be included in Fig. 4.

Response: We have added scale bars in Fig. 4a and f.

Reviewer #2

The article by Schöneberg et al represents an interesting attempt to recover, computationally, and by super-resolution imaging the sequence of the protein organization events leading to formation of endocytic neck. According to the major idea of the model, the background for the process includes enzymatic conversion of [PI(4,5)P₂] into [PI(3,4)P₂] underneath the clathrin coat, [PI(3,4)P₂] lateral diffusion into the membrane region surrounding of the coat, and recruitment of BAR domain protein SNX9. The model predicts a localized SNX9 binding along the rim of the coat. The model suggests that formation of a ring-like region enriched in SNX9 around the coat generates additional bending of the endocytic neck and assists dynamin in driving the neck fission.

The study is very systematic, thorough and represents a great example of a concerned computational-experimental study, where experimental knowledge determines the values of parameters of the theoretical model, while the predictions following from the model solution are checked by specifically designed experiments. I think the article deserves publication in Nature Communications after some revision of the presentation.

Response: We thank the referee for these very positive remarks and for endorsing publication of our work in *Nature Communications*.

1. It will help if the authors describe in simple terms and straightforwardly the scenario predicted computationally and explain the physics behind it. Such description may already exist in the paper, but it is spread over the text and is, therefore, difficult to summarize. Specific questions would be:

(i) why does the model predict concentration of SNX9 within a limited region near the coat edge rather than SNX9 spreading all over the membrane outside the coat? If this is due to SNX9 oligomerization, which builds around the coat a fence impermeable for diffusion of newly synthesized SNX9 molecules, what happens with the latter? How is the steady state reached in this case?

Response: We have clarified this question in the revised version. Increased and localized concentration of SNX9 depends on at least the following factors: (i) Increased PI(3,4)P₂ concentration, (ii) availability of space to dock on the membrane, (iii) protein-protein interactions, and, (iv) availability of a suitable membrane curvature.

While the simulations in the previous manuscript described PI lipid diffusion in a bounded simulation cell, our new simulations use absorbing boundaries (i.e. lipids are removed when they hit the boundary). We show that in this case, a PI(3,4)P₂ gradient sufficient to recruit SNX9 can only be obtained if lipid concentration is slowed down by two orders of magnitude compared to diffusion if plain membranes. Such a slowdown of lipid diffusion under protein coats has indeed been observed experimentally (Zhao et al., Cell Rep. 2013).

The preferred localization of SNX9 at the neck is mainly due to two effects: The neck has more available space than the CCP coat, where PI(3,4)P₂ is produced, and at the neck, SNX9 can engage in oligomeric interactions that further stabilize its position there. While both effects would also be expected to be present outside the CCP, the PI(3,4)P₂ concentration will decay with increasing distance from the CCP, and the membrane is flat outside the CCP, which is not the preferred template for SNX9. Consistent with that, our STORM experiments do not show

SNX9 to be enriched at flat membranes outside CCPs. In the revised manuscript we have added the following conclusion to the section "*Timed recruitment of SNX9 to the endocytic vesicle neck*":

"These results suggest that timed lipid-driven recruitment of SNX9 causes its localization and concentration at the endocytic vesicle neck, possibly at the interface with the clathrin coat. While PI(3,4)P₂ is produced under the protein-dense clathrin coat, SNX9 assembles at the neck where it finds sufficient space, oligomeric contacts to engage in, and, presumably, also a preferred membrane curvature. Consistently, the SNX9 concentration is not found to be increased at the flat membrane around the CCP, although this area is expected to contain elevated levels of PI(3,4)P₂."

(ii) Is the time sequence of protein recruitment – SNX9 before dynamin - the result of the model or an assumption?

Response: The time course of PI3KC2 α and SNX9 to CCPs has been analyzed in our earlier paper (Posor et al., Nature 2013) and confirmed in genome-engineered SK-MEL-2 cells co-expressing mRFP-clathrin light chain (mRFP-CLC) together with dynamin 2-eGFP from their endogenous loci. These data are reported in Figure 4 of the present Ms.

2. I got an impression, that the computations were performed for a flat rather than curved membrane.

Response: Indeed the protein recruitment simulations (Fig. 2) are performed on a flat membrane, which is now explicitly mentioned in the text. We have chosen to do this for two reasons: (i) Computational simplicity - in principle the topology of the reaction-diffusion simulation would need to change over time as the U-shaped membrane transforms into an Ω -shaped membrane, but the exact way how this transition occurs is still speculative, and (ii) a flat membrane is the more conservative option. If a curved membrane simulation would be used, it would have a smaller boundary, which would additionally reduce the flux of produced PI(3,4)P₂ lipids outside the CCP and, thus, increase their local concentration, further enhancing the observed effects.

In summary, the 2D recruitment simulations are an *idealized setup* to study the ability of PI lipid production to serve as a switch for SNX9 protein recruitment. The three-dimensional localization of the recruited proteins is then studied in the 3D particle diffusion simulations described in Fig. 4. Note that the latter simulations can currently not be extended to the simulation time of roughly one minute required for the 2D recruitment simulations.

If this is the case, it should be discussed how the shape of a real membrane neck can affect diffusion of the membrane-bound proteins and whether this may influence the model results.

Response: We have now added the following sentences in the first results section, after discussing the effects of different PI-lipid diffusion constants:

"Note that our protein recruitment simulations are conducted on a flat membrane, which also corresponds to a conservative setting. Endocytic intermediates are U or Ω -shaped, which involves an additional diffusion bottleneck around the clathrin coat that reduces the flux of produced $PI(3,4)P_2$ outside the CCP, and thus enhances the effects described here."

3. The suggestion of the model, that SNX9 concentration can shape the membrane neck to the extent implied by the experimental results should be supported, quantitatively, by structural features of this specific protein. Indeed, the curvatures produced and sensed by different BAR domains vary in a broad range from about 1/10nm for endophilin and amphiphysin, through very low values for F-BARs, to negative values for I-BARs. Hence, to justify the model a number is needed for SNX9.

Response: In order to use the curvature of SNX9 as an argument for the membrane tubes it would fit, one would additionally need to know the angle of SNX9 dimers with respect to the membrane neck equator. In the revised manuscript we have therefore argued as follows:

We have first determined the average peak number of SNX9 copies recruited to CCPs during successful endocytosis events, which have been measured using new TIRF experiments conducted in living cells (see our response to referee 3). These results suggest that 40 SNX9 molecules are sufficient to drive endocytosis.

Placing these 40 SNX9 molecules onto CCPs with different neck diameters as shown in the **new Figure 4e** suggests that accumulation of SNX9 at the pit increases curvature in a gradual process that ends, when the PX-BAR is able to form a complete ring around the membrane template. This is achieved at a membrane diameter of 40 nm, which is consistent with the fact that prior experimental work suggests that SNX9 via its PX-BAR domain preferentially shapes membranes into 40 nm diameter tubules in vitro (Yarar et al., 2007). If constriction proceeds further, SNX9 would assemble in double rings and form a 20 nm diameter tube, which is compatible with known membrane fission templates for dynamin (Dar et al, 2015).

4. Since the authors conclude that SNX9 promotes the fission process, some thoughts are needed on a specific mechanism of this effect. If no reasonable idea could be suggested at this stage, I suggest to downplay in the paper the issue of membrane fission.

Response: The major finding reported in our manuscript is that $PI(3,4)P_2$ -recruited SNX9 promotes membrane constriction prior to fission by dynamin. This may be largely for geometrical reasons as dynamin-mediated fission is greatly facilitated by highly curved membrane templates and SNX9/18 are found to contribute to the formation of such templates in living cells as overt from the accumulation of U-shaped endocytic intermediates (see Figure 6).

While our paper provides multiple lines of evidence that the $PI(3,4)P_2$ - SNX9 coupling is the primary source of membrane constriction, we do not provide the molecular mechanism by which SNX9 constricts the membrane. This has been explicitly said in the revised manuscript text.

Reviewer #3

The manuscript by Schöneberg et al. is interested in the specific recruitment of Snx9 to the neck of endocytic pits by the specific production of PI(3,4)P2 by the enzyme PI3KC2. It shows that, even if the PX domain of Snx9 has no preference for one of its 2 possible ligands PI(3,4)P2 and PI(4,5)P2, it is specifically recruited to the pit by PI(3,4)P2 because of competition of other CCPs players also binding PI(4,5)P2. The manuscript is a nice combination of in silico simulations, biochemical measurements to extract relevant parameters, and cell biology experiment to qualitatively validate the model.

Response: We thank the referee for these very positive remarks and for highlighting the importance and high quality of our work.

1-Diffusion of PI(3,4)P2, enzymatic activity of PI3KC2 and time recruitment of Snx9

From their simulation, the authors show that the recruitment of Snx9 is perfectly coordinated with the increase of PI(3,4)P2 in the membrane. However in cells, as shown by the authors in their previous paper (Posor et al.), PI3KC2 comes much before Snx9, suggesting that the Snx9 comes long after the start of PI(3,4)P2 production. This discrepancy has to be solved, or explained to support the conclusion of the authors that Snx9 is recruited to the pit by PI(3,4)P2.

In my opinion, it could come from a diffusion problem. When produced by PI3KC2, the PI(3,4)P2 would diffuse very fast out of the pit, not increasing locally its concentration. In this case, it would only be when PI3KC2 activity at the pit reaches a certain threshold, enough to overcome the lipid diffusion, that the local concentration of PI(3,4)P2 would increase, recruiting Snx9. But what is the enzymatic activity of PI3KC2 at the pit?

The authors state that from their enzymatic activity measurement, and the average number of PI3KC2 subunits present in the coat, they can compute the PI(3,4)P2 production. But they don't show any number nor detailed explanation. Going back to Borner et al. JCB 2012, I found that on average, there are 10 copies of PI3KC2 per CCV. Using the enzymatic activity measured by the authors (4 per s), this leads to the production of about 40 PI(3,4)P2 per second. How fast do these 40 lipids diffuses?

The authors simulate 0.12 μm^2 of membrane, which would take about 0.04 s for each lipid to move out this area, since they diffuse (using the value given by the authors) at 3 $\mu\text{m}^2/\text{s}$. That means that on average, there is $40 \times 0.04 = 1.6$ PI(3,4)P2 present in the entire membrane area simulated by the authors with the average enzymatic activity of PI3KC2. Thus diffusion of lipids is clearly limiting, and I request the authors to perform the following simulations/experiments/explanations to address this point.

First, it is not clearly indicated in the paper (I could not find a clear indication), but I guess the authors performed the simulation with a closed area of 0.12 μm^2 , meaning that each molecule added to this area would diffuse fast, but stay in the boundaries of the area. If this is the case, I would suggest that simulations should be run with a change of boundary conditions, in which PIP lipids (only), when they pass the boundary of the simulated domain, are lost, to mimic a case in which a CCP is connected to an infinite membrane, a case more physiologically relevant. This will allow to find the sufficient PI(3,4)P2 production rate to overcome diffusion.

Response: We thank the referee for raising this very important point that we have tackled in the following way: We have conducted new simulations that account for the possibility of lipid diffusion using absorbing boundary conditions as suggested by referee 3. Using this model together with experimental data (Zhao et al., Cell Rep. 2013) on the restriction of lipid diffusion by endocytic BAR domain proteins (also referred to by referee 1) we now show in the **new Figure 2** that clathrin-coated pit enrichment of PI(3,4)P₂ synthesized by PI3KC2 α depends on the restriction of lateral lipid diffusion by endocytic proteins assembled underneath the clathrin coat. Our new simulations confirm the prediction of referee 3 that the buildup of a local PI(3,4)P₂ gradient at coated pits depends on the ability of the endocytic coat to limit the lateral diffusion of PI lipids by at least two orders of magnitude, in agreement with recent experimental data (Zhao et al., Cell Rep. 2013). These results provide strong additional support to our initial findings and considerably extend the computational model. Moreover, they confirm - as predicted by the referee - that bound endocytic proteins must slow lipid diffusion for the system to work.

Second, the calculation I did above is pretty gross. In my opinion, as the authors have shown in their previous paper that PI3KC2 accumulate with time in the CCP, I could envision a mechanism in which, at a given time during the accumulation of PI3KC2 at the pit, the production rate of PI(3,4)P₂ overcomes the diffusion and starts accumulating PI(3,4)P₂ at the pits to recruit Snx9. In this case, it would be expected the recruitment of Snx9 starts way after the accumulation of PI3KC2, as seen in cells. I would thus request the authors to perform simulations with the same PIP losing condition as above, but with the accumulation of PI3KC2 and/or steady increase of PIP₃,4P₂ production. Overall, I think it would be essential to model the PI3KC2 activity and diffusion of PIPs to solve the discrepancy.

Response: Please note that our simulations cannot describe the buildup of the clathrin coat and there is not sufficient experimental information to do that quantitatively. Therefore, we start with an assembled clathrin coat and a fixed copy number of PI3KC2 α . To this end, we have now determined the absolute copy number of PI3KC2 α recruited to clathrin-coated pits using quantitative TIRF imaging (see below), and used this number in combination with the new computational simulations that allow for lipid diffusion to assess the endocytic protein dynamics (see also our response to the previous point).

As seen in the **new Fig. 2** we still observe SNX9 copy numbers to rise roughly in parallel to the accumulation of PI(3,4)P₂. Note that a delay of the response could simply be achieved by reducing the protein-ligand binding and dissociation rates, which would not change the binding affinities. Since the absolute value of the binding and dissociation rates are not known, this setting would be arbitrary, and we, thus, prefer not to make predictions of the absolute binding dynamics. The results reported in Fig. 2 focus on the investigation of binding mechanisms, i.e. under which circumstances is a robust recruitment of SNX9 observed.

In cells, it would be interesting to calibrate the microscope to estimate the number of PI3KC2 copies per pit with time. This would allow for comparing with the simulations proposed above, and show that the overall production rate of PI₃,4P₂ is sufficient for overcoming the lipid diffusion.

Response: Again, an important point that we have addressed experimentally: We have determined the absolute copy number of PI3KC2 α recruited to clathrin-coated pits using quantitative TIRF imaging: We used genome-engineered SK-MEL-2 cells expressing mRFP-clathrin light chain (mRFP-CLC) from its endogenous locus (Grassart et al J Cell Biol 2014). Previous data using genome-engineered SK-MEL-2 cells have shown that peaks of dynamin 2-GFP recruitment correspond to an average recruitment of about 35 dynamin molecules per CCP (Grassart et al J Cell Biol 2014) (**new Fig. 1e-g**). We confirmed that a similar number of dynamin 2 molecules is recruited to CCPs upon plasmid-based re-expression of siRNA-resistant dynamin 2-GFP (Faelber et al., Nature 2011) in genome-engineered cells depleted of endogenous dynamin 2 (**new Fig. 1h**), validating our approach. Expression of GFP-PI3KC2 α in SK-MEL-2 cells depleted of endogenous PI3KC2 α revealed an average recruitment of about 33 PI3KC2 α molecules per CCP (shown in the **new Figure 1e-h**), consistent with quantitative proteomic data by Borner et al (2012).

2-the 100 protein threshold.

I have a problem with the “constriction threshold” that the authors have chosen. This is essential because if the value of the threshold (100 copies) is different, all the conclusions of the authors are likely to be wrong. For example, a threshold of 50 copies would completely change the conclusions of figure 2. The rationale for choosing 100 as the constriction threshold seems fairly arbitrary to me. Based on structural arguments, the authors say it is the minimal number of copies to form a full ring around U-shaped pits. However, constriction and membrane deformation (see also point 3) by BAR proteins is a continuous process, and depends on the membrane-bound densities, not on a specific number (see Sorre et al. PNAS 2012, and Simunovic et al. PNAS 2016). Thus constriction will be happening as soon as a single copy will be present. The curvature generated by 80 copies is likely not to be so much different from the curvature generated by 200 copies, which are presented as non-constricted (early stage below the threshold) and constricted state of the neck (late stage above the threshold). Thus, the conclusions of the authors that the accumulation of PI(3,4)P2 drives a further recruitment of Snx9 above a threshold allowing for constriction seems strongly dependent on the threshold value, is, in my opinion, conceptually wrong. I would instead propose that there is a continuous accumulation of Snx9 at the pit, which may increase curvature, in conjunction with other BAR proteins (see below).

In order to define more rigorously the threshold value, I request the authors to quantify the number of Snx9 copies at the pit by calibrating their microscope. In this case, the threshold value could be set as 50% of the maximal number of copies at the pit. It would be less arbitrary.

Response: This is indeed an excellent point that has motivated us to carry out a large number of additional experiments:

First, we have determined the absolute copy number of SNX9 recruited to clathrin-coated pits using quantitative TIRF imaging: We used genome-engineered SK-MEL-2 cells expressing mRFP-clathrin light chain (mRFP-CLC) from its endogenous locus (Grassart et al J Cell Biol 2014). Previous data using genome-engineered SK-MEL-2 cells endogenously expressing dynamin 2-GFP have shown that peaks of dynamin 2-GFP recruitment correspond to an average recruitment of about 35 dynamin molecules per CCP (Grassart et al J Cell Biol 2014) (**new Fig. 1e-g**). We confirmed that a similar number of dynamin 2 molecules is recruited to CCPs upon

plasmid-based re-expression of siRNA-resistant dynamin 2-GFP (Faelber et al., Nature 2011) in genome-engineered cells depleted of endogenous dynamin 2 (**new Fig. 1h**), validating our approach. Analysis of SK-MEL-2 cells depleted of endogenous SNX9 and its close paralog SNX18 and functionally rescued by re-expression of iRFP-SNX9 yielded an average peak recruitment of 41 SNX9 molecules (based on monomers) per CCP (shown in the **new Figure 1e-h**).

Second, we have modeled the number SNX9 molecules (monomers) based on the crystallographically determined size and shape of its PX-BAR domain. We calculate that about 40 SNX9 molecules are required to assemble a full PX-BAR domain ring around a 40 nm-wide membrane neck formed by self-assembled SNX9 molecules on lipid templates *in vitro* (Yarar et al., 2007). Given the close match of experimentally determined SNX9 copy numbers recruited to CCPs in living cells and these theoretical considerations based on the self-assembly of its PX-BAR domain we regard 40 SNX9 molecules as sufficient to drive constriction. In the revised manuscript, we therefore consider SNX9 recruitment to be successful if the SNX9 copy number is increased from below 30 to over 40 molecules (grey shaded area shown in the new Figure 2).

Third, we have placed these molecules onto the neck of clathrin-coated pits of different sizes in the **new Figure 4e**. These simulations are highly compatible with the referee's prediction that accumulation of SNX9 at the pit increases curvature in a gradual process until the PX-BAR is able to form a complete ring around the membrane template. It is possible, but currently hypothetical that the process continues towards a double-ring and an even narrower neck.

*Finally, the simulations seem to indicate that the presence of PI(4,5)P₂ is sufficient to drive Snx9 binding in absence of PI(3,4)P₂ (and in presence of competing PI(4,5)P₂ binding partners). I would thus have expected Snx9 to be at least a little bit present from the very start of the CCP formation. However, in cells, Snx9 seems to be present only at the end of the CCP formation, whereas all other partners (PI(4,5)P₂ competing components of the clathrin coat) are present from the beginning. It seems inconsistent to me, and I thus wonder how much the simulations reflect the *in vivo* situation.*

Response: Our simulation essentially reflects a situation halfway through the endocytic process when clathrin coats have fully assembled, e.g. a situation about 50 s prior to fission when PI3KC2 α is recruited in Figure 4b. Our live imaging data clearly are limited with respect to sensitivity and, in our view are only grossly compatible with the simulations. However, as not all protein-protein interactions and geometries are reflected in the computational model not all protein stoichiometries will be accurately simulated and thus, it is possible that SNX9 copy numbers prior to PI(3,4)P₂ synthesis are affected by additional factors (e.g. curvature as pointed out by the referee). We hope that the referee agrees that a complete modeling of all parameters relevant to endocytosis such as all additional protein interactions, membrane curvature, local differences in membrane tension imposed by actin etc. would require a vast number of additional experimental data that is currently not available, and are thus beyond the scope of the present study. We hope, however, that our approach will fuel future studies aimed at a more complete model of the *in vivo* situation. In the revised manuscript, we have added a brief discussion of the effects that have been neglected in the present simulations -- see also response to the next question.

3-constriction mechanism, role of other BAR proteins.

In all simulations, membrane curvature is not coupled to binding (Fig2 is made with flat membranes, and other figures do not allow for curvature change as a function of protein density). For BAR proteins, it is very well described that binding to the membrane is coupled to curvature. This effect is probably as important as the lipid binding effect, and should be taken into account. For example, the fact that Snx9 does not bind to the pit in early stage may not be due to the absence of PI(3,4)P₂, which is unlikely since PI3KC2 is present, but because the pit is not curved enough, not allowing for Snx9 BAR binding. Also, the ring organization discussed in fig4 and supp. figS2 strongly depend on the size of the neck, and thus on this coupling mechanism between binding and curvature. I understand it is not easy to simulate this coupling and that the algorithm written by the authors are not meant to do this, but I think it would be a plus if the authors could at least give us an idea of how the conclusions of figure 4 change with gradual increase of the neck size (only 2 extreme sizes have been tested).

Response: We agree that membrane-binding proteins sense and induce curvature. Unfortunately, a quantitative model would require a membrane mechanics model that can describe the coupling of local and global curvature and curvature-dependent interaction potentials between proteins and membranes. These in turn would require extensive biophysical membrane binding data for all proteins involved. While we are working on all these fronts, establishing a model that faithfully describes these effects is still years away.

As suggested by the reviewer, we have discussed the effects of neglected membrane curvature aspects in the revision. We are confident that our models are probing a conservative scenario. In our discussion of Fig. 2 we have added the following statement:

"Note that our protein recruitment simulations are conducted on a flat membrane, which corresponds to a conservative setting. Endocytic intermediates are U or Ω-shaped, which involves an additional diffusion bottleneck outside the clathrin coat that reduces the flux of produced PI(3,4)P₂ outside the CCP, and thus enhances the effects described here."

This effect is stronger for a Ω-shape than for a U-shape, which would add a nonlinear amplification of PI(3,4)P₂ on SNX9 concentration. In Fig. 4, the localization of SNX9 at the neck is likely to be further promoted by the membrane curvature at the neck. In the discussion of its results, we have added:

" While PI(3,4)P₂ is produced under the protein-dense clathrin coat, SNX9 assembles at the neck where it finds sufficient space, oligomeric contacts to engage in, and, presumably, also a preferred membrane curvature. Consistently, the SNX9 concentration is not found to be increased at the flat membrane around the CCP, although this area is expected to contain elevated levels of PI(3,4)P₂."

Additionally, we have now modeled the localization of SNX9 to various intermediate neck diameters (revised Fig. 4e).

Also, the structure of the BAR domain of Snx9 is very similar to the one of Endophilin and Amphiphysin, which appear at the pit at the same time than Snx9, but however do not bind PI3,4P2 specifically. I would have expected that if Snx9 is not present (siRNA), Endo and Amphi would totally compensate for the loss of Snx9. The authors should check the localization of Endo and Amphi in Snx9 siRNA cells, as well as in PI3KC2 siRNA cells. The accumulation of U-shaped CCPs in those cells suggest that both Endo and Amphi are not recruited, which make it difficult to believe that the non-constricted neck phenotype is solely due to the absence of Snx9...

Response: In response to this interesting question we have analyzed the expression levels and localization of amphiphysin 1 and endophilin A2 in control and SNX9/18-depleted cells. In our **new Suppl. Fig. 5** we report that amphiphysin 1 levels at coated pits are increased in SNX9/18-depleted cells, possibly as a result of compensation that, however, is insufficient to fully rescue loss of SNX9/18 as U-shaped clathrin-coated pits accumulate in these cells (please see Figure 6). We do not see any change in the (limited) colocalization of endophilin A2 with clathrin, confirming that its expression or recruitment to endocytic sites is not affected by loss of SNX9/18. These data also make it very unlikely that the functional defects in membrane constriction observed in SNX9/18-depleted cells are an indirect consequence of loss of endophilin or amphiphysin.

Reviewers' Comments:

Reviewer #1:

Remarks to the Author:

The authors have satisfactorily addressed my previous concerns.

Reviewer #2:

Remarks to the Author:

The authors thoughtfully addressed my concerns so that I recommend publication of the revised version.

Reviewer #3:

Remarks to the Author:

The authors have beautifully addressed all my (tough) comments, and I thus strongly recommend publication without further delays.